# On the One-sided Convergence of Adam-type Algorithms in Non-convex Non-concave Min-max Optimization

## Abstract

Adam-type methods, the extension of adaptive gradient methods, have shown great performance in the training of both supervised and unsupervised machine learning models. In particular, Adam-type optimizers have been widely used empirically as the default tool for training generative adversarial networks (GANs). On the theory side, however, despite the existence of theoretical results showing the efficiency of Adam-type methods in minimization problems, the reason of their wonderful performance still remains absent in GAN's training. In existing works, the fast convergence has long been considered as one of the most important reasons and multiple works have been proposed to give a theoretical guarantee of the convergence to a critical point of min-max optimization algorithms under certain assumptions. In this paper, we firstly argue empirically that in GAN's training, Adam does not converge to a critical point even upon successful training: Only the generator is converging while the discriminator's gradient norm remains high throughout the training. We name this one-sided convergence. Then we bridge the gap between experiments and theory by showing that Adam-type algorithms provably converge to a one-sided first order stationary points in min-max optimization problems under the one-sided MVI condition. We also empirically verify that such one-sided MVI condition is satisfied for standard GANs after trained over standard data sets. To the best of our knowledge, this is the very first result which provides an empirical observation and a strict theoretical guarantee on the one-sided convergence of Adam-type algorithms in min-max optimization.

## 1 Introduction

As one of the most popular optimizers in supervised deep learning tasks like natural language processing (Chowdhury, 2003) as well as the main workhorse of generative adversarial network training (Goodfellow et al., 2014), Adam-type methods are widely used because of their minimal need for learning rate tuning and their coordinate-wise adaptivity on local geometry. Starting from AdaGrad (Duchi et al., 2011), adaptive gradient methods have evolved into a variety of different Adam-type algorithms, such as Adam (Kingma & Ba, 2015), RMSprop, AMSGrad (Reddi et al., 2018) and AdaDelta (Zeiler, 2012). In supervised learning, adaptive gradient methods and Adam-type algorithms play important roles. Especially in the field of NLP (natural language processing), Adam-type algorithms are the goto optimizer. Multiple NLP experiments show that sparse Adam outperforms other non-adaptive algorithms like Stochastic Gradient Descent (SGD) not only on the solution performance, but also on both the training and testing error's convergence rates. It's worth mentioned that the most popular pre-training language model BERT (Devlin et al., 2018) also uses Adam as its optimizer, which shows the power of Adam-type algorithms.

Also, Adam-type algorithms are very effective in min-max optimization. As a direct and widely used application of min-max optimization, generative adversarial networks (GANs) are notorious for the training difficulty. Training by SGD will easily diverge to nowhere or converge to a limiting cycle, both of which will lead to an ill-performing solution, while Adam optimizer, as the default optimizer for GANs (Hsieh et al., 2020), can obtain better performance. The reason why these two optimizers have so much difference in GAN's training has long been an open problem. Traditionally, the training performance of min-max optimization is measured according to its first-order convergence, which means the norm of the gradient, but is it really true in GAN's training?

After training GAN on two relatively simple datasets, MNIST and Fashion-MNIST, we can find that, in a practical training process of GAN, Adam optimizer does not perfectly converge since the

norm of discriminator's gradient remains quite high through out the training process. Instead, it only has a one-sided convergence as the norm of generator's gradient actually converges to 0. This

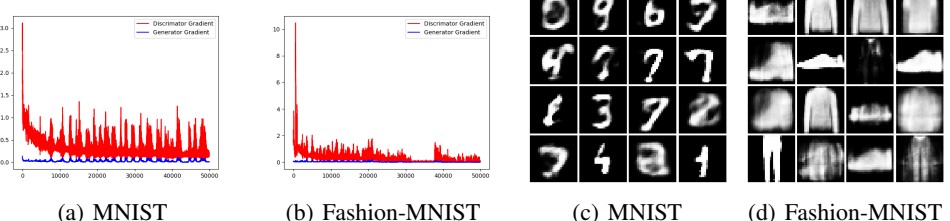

|  (a) MNIST | (b) Fashion-MNIST | (c) MNIST | (d) Fashion-MNIST |

Figure 1: We train GAN on the dataset MNIST and Fashion-MNIST. The first two figures above show us the Frobenius norm of the gradients of discriminator and generator. After 50k iterations, we obtain (c),(d) by using Adam. Despite its one-sided convergence, the min-max training actually succeeds.

paper thus aims to explain this phenomenon by bridging the gap between theory and practice. On one hand, we understand under which conditions Adam-type optimization algorithms have *provable convergence* for min-max optimization. Towards this end, a recent work (Liu et al., 2020) designs two algorithms, Optimistic Stochastic Gradient (OSG) and Optimistic AdaGrad (OAdaGrad) for solving a class of non-convex non-concave min-max problems and gives theoretical guarantee on their convergence. (Liu et al., 2020) also proposes an open problem on the convergence proof of Adam-type algorithms, which is solved by this paper. On the other hand, we find that the MVI condition needed for our convergence proof does not practically hold for GANs. Instead, we propose the much milder one-sided MVI condition, which tends to hold practically and under which we provide the theoretical guarantee of the one-sided convergence of Adam-type algorithms.

Despite some theoretical guarantee made on the convergence of Adam-type algorithms on convex concave or non-convex concave min-max optimization, in the non-convex non-concave setting which is most general, there is no theoretical guarantee on convergence. Comparatively speaking, proving the convergence of Adam-type algorithms is much more difficult since they use an empirical version of Momentum. Although it has been shown to perform well in practice, it is actually difficult to analyze theoretically. Even in the standard convex setting, proving the convergence of Adam-type algorithms (Reddi et al., 2018; Zou et al., 2021) is much harder than other adaptive algorithms such as AdaGrad (Duchi et al., 2011). Actually, the original version of Adam is known not to converge in convex settings. Therefore, to formally analyze the convergence of Adam-type algorithms in min-max optimization, we also consider a "theoretically correct" version of Adam, which is an analog of AMSGrad (Reddi et al., 2018).

In this paper, there are three main contributions. (1) We analyze Extra Gradient AMSGrad, which is an Adam-type algorithm used for solving non-convex non-concave min-max optimization problems as well as GAN's training. We prove that, under the assumption of standard MVI condition, the Extra Gradient AMSGrad algorithm provably converges to a $\varepsilon$-stationary point with $O(d\varepsilon^{-2})$ complexity in deterministic setting and $O(d\varepsilon^{-4})$ complexity in stochastic setting. (2) Although the standard MVI condition above is a much milder assumption than convexity, we empirically show that MVI condition does not hold for GAN's objective functions in reality. Instead, the one-sided MVI condition proposed by us tends to hold, which is the mildest assumption ever used in all the convergence proofs for min-max optimization. Under the the one-sided MVI condition, we modify the algorithm above by using dual rate decay, and theoretically prove its convergence rate. (3) We conduct empirical experiments on GAN's training by the Extra Gradient AMSGrad algorithm and the Extra Gradient AMSGrad with dual rate decay analyzed by us. We show that they have much better performance than the Stochastic Gradient Descent Ascent (SGDA) algorithm. Also, we empirically verify that **our new one-sided MVI condition is indeed satisfied during GAN's training while the previously proposed standard MVI condition is not**, which makes the one-sided MVI condition much closer to reality than the standard version.

After achieving all these results, we are eventually able to understand the one-sided convergence of Adam-type algorithms in min-max optimization as well as in GAN's training.

## 2 BACKGROUND AND RELATED WORKS

In this section, we will introduce the background knowledge as well as related works on the following three fields: adaptive gradient methods, min-max optimization, and the convergence properties of multiple algorithms for min-max optimization problems.

### 2.1 ADAPTIVE GRADIENT METHODS AND ADAM-TYPE METHODS

We consider the simplest 1-dimensional unconstrained minimization problem:

$$\min_{x \in \mathcal{D} \subseteq \mathbb{R}} f(x).$$

where $f : \mathcal{D} \to \mathbb{R}$ is a continuously differentiable function. As one of the most dominant algorithms on the optimization problem above, Stochastic Gradient Descent (SGD) was originally proposed by (Goodfellow et al., 2016), which has been both empirically and theoretically proved effective, especially when facing large datasets and complicated models. To further improve the performance of SGD, several adaptive variants of SGD have been proposed, such as RMSprop, Adam (Kingma & Ba, 2015), AdaGrad (Duchi et al., 2011), AMSGrad (Reddi et al., 2018) and AdaDelta (Zeiler, 2012). Distinguished from the vanilla gradient descent or its stochastic version SGD, adaptive gradient methods use a coordinate-wise scaling of the updating direction and each iteration relies on the history information of past gradients. In AdaGrad, we use arithmetic average when adopting history gradient information of each iteration while in Adam, RMSprop etc., we use exponential moving average instead because its believed that the more current gradient information is more important. Although adaptive gradient methods and momentum based methods are two different routes on optimization, they are combined perfectly in Adam. Now we introduce the family of adaptive gradient methods and Adam-type, and all of them have the following form:

$$m_{t+1} = h_t \nabla f(x_t) + r_t \cdot m_t, \ v_{t+1} = p_t (\nabla f(x_t))^2 + q_t \cdot v_t$$
$$x_{t+1} = x_t - \lambda_t \cdot \frac{m_{t+1}}{\sqrt{v_{t+1}} + \varepsilon}. \qquad \text{[Adaptive]}$$

Here, $f$ is the objective function to minimize. $h, r, p, q$ are scalars depending on $t$, $\lambda_t$ is the learning rate of the $t$-th iteration and $\varepsilon > 0$ is a small constant used to protect the denominator from being close to 0. From the formula above, we see that the momentum $m_t$ is the weighted sum of the past gradients and $v_t$ is the weighted sum of the past squared gradients. When $h = 1, r = 0$, $m_{t+1} = \nabla f(x_t)$ is just the current gradient. We start with the original Adam.

$$v_{t+1} = \alpha_t v_t + (1 - \alpha_t)(\nabla f(x_t))^2, \ m_{t+1} = \beta_t m_t + (1 - \beta_t) \nabla f(x_t)$$
$$x_{t+1} = x_t - \lambda \cdot \frac{m_{t+1}}{\sqrt{v_{t+1}} + \varepsilon}. \qquad \text{[Adam]}$$

As we can see, Adam is a combination of adaptive gradient method and momentum method. Here, the momentum term is empirical, meaning that it does not coincide with acceleration techniques that are theoretically sound, which creates extra difficult for the analysis. In Adam, we have $h_t + r_t = p_t + q_t = 1$. When the $\alpha_t = \alpha, \beta_t = \beta$ remains constant, there is a bias correction step where $v_{t+1} \leftarrow \frac{v_{t+1}}{1-\alpha^t}$ and $m_{t+1} \leftarrow \frac{m_{t+1}}{1-\beta^t}$. However, we may practically ignore this bias correction step since $\frac{1}{1-\alpha^t}$ and $\frac{1}{1-\beta^t}$ rapidly approach to 1. As one of the variants of Adam, AMSGrad has the following formulation:

$$\hat{v}_{t+1} = \alpha_t v_t + (1 - \alpha_t)(\nabla f(x_t))^2, \ v_{t+1} = \max(v_t, \hat{v}_{t+1})$$
$$m_{t+1} = \beta_t v_t + (1 - \beta_t) \nabla f(x_t), \quad x_{t+1} = x_t - \lambda \cdot \frac{m_{t+1}}{\sqrt{v_{t+1}} + \varepsilon}. \qquad \text{[AMSGrad]}$$

As we can see, their difference is that the velocity term $v_t$ keeps increasing in AMSGrad.

After showing the details of these traditional adaptive gradient methods and Adam-type methods, we introduce their convergence properties as well as their further variants. Reddi et al. (2018) shows that Adam does not converge in some settings where large gradient information is rarely encountered and it will die out quickly because of the "short memory" property of the exponential moving average. However, under some conditions, the convergence proofs of adaptive gradient methods have been obtained. Basu et al. (2018) proved the convergence rate of RMSprop and Adam when using deterministic gradients instead of stochastic gradients. Li & Orabona (2018) analyzed the

convergence rate of AdaGrad under both convex and non-convex settings. All the papers above provide theoretical guarantee for the convergence of different types of adaptive gradient descent. After that, Chen et al. (2019) extends Adam to a broader class of Adam-type algorithms and provides its convergence analysis for non-convex optimization problems. In order to combine the fast convergence of adaptive methods and better generalization with momentum based methods, a number of new algorithms are proposed, such as SC-AdaGrad / SC-RMSprop (Mukkamala & Hein, 2017), AdamW (Loshchilov & Hutter, 2019), AdaBound (Luo et al., 2019) etc..

## 2.2 MIN-MAX OPTIMIZATION

In the min-max optimization problem (or saddle point problem), we have to solve:

$$\min_{x \in \mathcal{X}} \max_{y \in \mathcal{Y}} \phi(x, y), \qquad \text{[SP]}$$

where $\mathcal{X} \subseteq \mathbb{R}^{n_1}, \mathcal{Y} \subseteq \mathbb{R}^{n_2}$, and $\phi : \mathcal{X} \times \mathcal{Y} \to \mathbb{R}$ is the objective function. When $\phi$ is convex on $x$ and concave on $y$, we call it a convex-concave min-max optimization. Otherwise, it's a more general non-convex non-concave min-max optimization. For the brevity, we denote $z = (x, y)$ and $\mathcal{Z} = \mathcal{X} \times \mathcal{Y} \subseteq \mathbb{R}^{n_1+n_2}$. We also introduce our gradient vector field: $V(z) = (-\nabla_x \phi(x, y), \nabla_y \phi(x, y))$, which are the update directions on both sides. The goal of [SP] is to find a tuple $z^* = (x^*, y^*)$ such that $\phi(x^*, y) \leqslant \phi(x^*, y^*) \leqslant \phi(x, y^*)$ holds for $\forall x \in \mathcal{X}, y \in \mathcal{Y}$, which is called the solution of [SP]. If the inequality above only holds in the local neighbourhood of $z^*$, then $z^*$ can only be called a local solution. Notice that the necessary condition of being a solution (or even a local solution) is to be a stationary point of $\phi$, which means $V(z^*) = 0$. Furthermore, if $V$ is $C^1$, any local solution of [SP] must be stable, which means $\nabla_{xx}^2 \phi(x^*, y^*) \succeq 0$ and $\nabla_{yy}^2 \phi(x^*, y^*) \preceq 0$. Next, we will introduce several commonly-used algorithms which are designed to solve [SP].

**Stochastic Gradient Descent Ascent (SGDA)** This is a simple extension of Stochastic Gradient Descent (SGD) algorithm for minimization problems (Johnsen, 1959). In the $t$-th iteration:

$$z_{t+1} = z_t + \gamma_t \cdot V(z_t; \omega_t), \qquad \text{[SGDA]}$$

where $\omega_1, \omega_2, \cdots$ are the independent and identically distributed sequence of noises. $V(z, \omega)$ can be treated as a query to the stochastic first-order oracle (SFO). In each iteration of SGDA, we need to query SFO once. Notice that, we simultaneously update $x, y$ in each iteration of SGDA. Therefore, if we alternate the updates of $x$ and $y$, we obtain a variant of SGDA, which is named as the alternating stochastic gradient descent ascent (AltSGDA) algorithm. Different from original SGDA, we have to make two queries to SFO in each iteration. One for $z_t = (x_t, y_t)$, and the other for the intermediate step $(x_{t+1}, y_t)$. Since original SGDA is not going to work even in the convex-concave setting (such as $\min_x \max_y f(x, y) = xy$), so researchers propose the following "theoretically correct modification".

**Stochastic Extra-gradient (SEG)** This is a different algorithm with the above SGDA, and it is originally proposed for solving the convex-concave setting of min-max optimization problems by Korpelevich (1976). Given $z_t$ as a base, we take a virtual gradient descent ascent step and obtain a $\tilde{z}_t$, which can be treated as the shadow of $z_t$. Then we use the gradient at $z_t'$ as the update direction of $z_t$. This process can be described as:

$$z_t' = z_t + \gamma_t \cdot V(z_t; \omega_t^{(1)})$$
$$z_{t+1} = z_t + \gamma_t \cdot V(z_t'; \omega_t^{(2)}). \qquad \text{[SEG]}$$

In each iteration, we need to make two queries to the SFO. One for the base $z_t$ and the other for the shadow $z_t'$. However, in the first step of [SEG], we can use the gradient at the previous shadow $z_{t-1}'$ so that we only have to make only one query in each iteration and remember the query's result of the previous step. This algorithm is called Optimistic Gradient or Popov's Extra-gradient (Popov, 1980) which can be described as:

$$z_t' = z_t + \gamma_t \cdot V(z_{t-1}'; \omega_{t-1})$$
$$z_{t+1} = z_t + \gamma_t \cdot V(z_t'; \omega_t). \qquad \text{[OG]}$$

As a widely used algorithm, it has been applied in multiple works (Daskalakis et al., 2018; Mertikopoulos et al., 2019). Under some mild assumptions, convergence rates are proved by many theoretical works and we will summarize them in the next section.

## 2.3 CONVERGENCE RATES OF MULTIPLE MIN-MAX ALGORITHMS

In this section, we summarize the convergence rates of different algorithms as well as the assumptions needed. For convex-concave optimization, Nesterov (2007) provided the $O(1/T)$ convergence guarantee of Mirror-Prox in terms of duality gap. Juditsky et al. (2011) introduced its stochastic version where only the stochastic first order oracle can be accessed. After combining with (Darzentas, 1983), convergence rates for both deterministic and stochastic mirror-prox algorithms are shown to be optimal. When it comes to the more challenging non-convex non-concave min-max optimization, Dang & Lan (2015) showed that the deterministic extragradient method can converge to $\varepsilon$-first order stationary point with non-asymptotic guarantee. Another interesting algorithm Inexact Proximal Point (IPP) method (Lin et al., 2018), which is a stage-wise algorithm, performs well when the objective function is weakly-convex weakly-concave. In each stage, we construct a strongly-convex strongly-concave sub-problem by adding quadratic regularizers. Then, by using stochastic algorithms, we can approximately solve the original problem. It's known that IPP also has a first order convergence guarantee. Also, Sanjabi et al. (2018) proposed an alternating deterministic optimization algorithm, where multiple steps of gradient ascents are conducted before one gradient descent step. Therefore, we can approximately make sure that the max step always reaches near optimal. However, in order to guarantee its convergence to first order stationary point, we have to assume that the inner maximization problem satisfies PL condition (Polyak, 1969). For the details of convergence rate, we summarize them into Table 1. Finally, MVI condition needs to be explained. Let $K : \mathbb{R}^d \to \mathbb{R}^d$ be

|  | Assumption | IC | Guarantee |
|---|---|---|---|
| OMD (Daskalakis et al., 2018) (deterministic) | bilinear | N/A | asymptotic |
| OG (Liu et al., 2020) (stochastic) | MVI has solution | $\mathcal{O}(\varepsilon^{-4})$ | $\varepsilon$-SP |
| OAdaGrad (Liu et al., 2020) (stochastic) | MVI has solution BCG Condition | $\widetilde{\mathcal{O}}\left((d/\varepsilon^2)^{\frac{1}{1-\alpha}}\right)$ | $\varepsilon$-SP |
| SEG (Iusem et al., 2017) (stochastic) | pseudo-monotonicity | $\mathcal{O}(\varepsilon^{-4})$ | $\varepsilon$-SP |
| Extra-gradient (Azizian et al., 2019) (deterministic) | strong-monotonicity | $\mathcal{O}(\log(1/\varepsilon))$ | $\varepsilon$-optim |
| AltSGDA(Gidel et al., 2019) (deterministic) | bilinear | $\mathcal{O}(\log(1/\varepsilon))$ | $\varepsilon$-optim |
| IPP (Lin et al., 2018) (stochastic) | MVI has solution | $\mathcal{O}(\varepsilon^{-6})$ | $\varepsilon$-SP |
| Extra Gradient AMSGrad (ours) (deterministic & stochastic) | MVI has solution | $\mathcal{O}(d\varepsilon^{-2})$ & $\mathcal{O}(d\varepsilon^{-4})$ | $\varepsilon$-SP |
| Extra Gradient AMSGrad with Dual Rate Decay (ours) (deterministic & stochastic) | **one-sided MVI has solution** | $\widetilde{\mathcal{O}}(d\varepsilon^{-2})$ & $\widetilde{\mathcal{O}}(d\varepsilon^{-4})$ | $\varepsilon$-SP |

Table 1: Summary of different algorithms for min-max optimization. IC stands for iteration complexity, $\varepsilon$-SP stands for $\varepsilon$-first order stationary point, and $\varepsilon$-optim stands for $\varepsilon$-close to the set of optimal solutions. The last two lines are algorithms analyzed by us in this paper. BCG condition stands for the bounded cumulative gradient assumption.

an operator and $\mathcal{X} \subseteq \mathbb{R}^d$ is a closed convex domain. Hartman & Stampacchia (1966) proposed the Stampacchia Variational Inequality (SVI), which aims to find $z^* \in \mathcal{X}$, such that $\langle K(z^*), z - z^* \rangle \geq 0$ holds for all $z \in \mathcal{X}$. Similarly, Minty (1962) proposed the Minty Variational Inequality (MVI) problem, which aims to find $z^* \in \mathcal{X}$, such that $\langle K(z), z - z^* \rangle \geq 0$. In Table 1, the operator $K(z) = (\nabla_x \phi(x,y), -\nabla_y \phi(x,y))^\top = -V(z)$ with $z = (x,y)$.

## 3 MAIN RESULTS

In this section, we introduce the main results of this paper. We focus on two algorithms: Extra Gradient AMSGrad (AMSGrad-EG) and Extra Gradient AMSGrad with Dual Rate Decay (AMSGrad-EG-DRD) which inherit the idea of OAdaGrad into Adam-type algorithms. With AMSGrad-EG, we can prove its first-order convergence under MVI condition. However, as we stated above, Adam does not perfectly converge in GAN's training since MVI condition does not always hold for GAN's objective functions. We bridge the gap by proposing one-sided MVI condition which is shown to be

more likely to hold. Under this condition, we prove that Extra Gradient AMSGrad with Dual Rate Decay (AMSGrad-EG-DRD) converges one-sidedly, which matches our experiment results.

## 3.1 PROBLEM SETTING AND ASSUMPTIONS

Throughout the paper, we analyze the min-max optimization problems:

$$\min_{x \in \mathcal{X}} \max_{y \in \mathcal{Y}} \phi(x, y), \qquad \text{[SP]}$$

where $\mathcal{X} \subseteq \mathbb{R}^{n_1}, \mathcal{Y} \subseteq \mathbb{R}^{n_2}$, and $\phi : \mathcal{X} \times \mathcal{Y} \to \mathbb{R}$ is the objective function. We denote $z = (x, y)$ and $\mathcal{Z} = \mathcal{X} \times \mathcal{Y}$. First, we state some useful assumptions on $\phi(x, y)$:

**Assumption 1.**
*(1) $V := (-\nabla_x \phi, \nabla_y \phi)$ is L-Lipschitz continuous under $\|\cdot\|_2$ norm.*
*(2) The stochastic first order gradient oracle (SFO) is unbiased and has bounded variance:*

$$\mathbb{E}[V(z; \xi)] = V(z) \ \text{ and } \ \mathbb{E}\|V(z; \xi) - V(z)\|^2 \leqslant \sigma^2.$$

*(3) The Stochastic first-order Gradient Oracle (SFO) has bounded output: there exists $G > 0$ and $\delta > 0$ such that $\|V(z; \xi)\|_2 \leqslant G$ and $\|V(z; \xi)\|_\infty \leqslant \delta$ almost surely holds.*
*(4) There exists a universal constant $D > 0$, such that $\|z_k\|_2 \leqslant D$ holds for all points $z_k$ on our trajectory and $\|z^*\|_2 \leqslant D$. If the feasible set $\mathcal{Z}$ is bounded, then this assumption naturally holds.*

**Assumption 2** (Standard MVI condition). *The MVI of $-V(z)$ has a solution, which means there exists a $z^*$, such that:*

$$\langle -V(z), z - z^* \rangle \geqslant 0 \text{ holds for } \forall z \in \mathcal{Z}$$

.

## 3.2 EXTRA GRADIENT AMSGRAD (AMSGRAD-EG)

In this section, we analyze the Extra Gradient AMSGrad (AMSGrad-EG) algorithm, which is used for non-convex non-concave min-max optimization, and we theoretically provide its convergence rate. So far, the convergence rate of Adam-type algorithms for min-max optimization has long been an open problem, and this work is the very first to obtain a related result. AMSGrad-EG algorithm is described as Algorithm 1.

---

**Algorithm 1** Extra Gradient AMSGrad

**Input:** The initial state $z_0 = m_0 = v_0 = 0$, a constant learning rate $\eta$, momentum parameters $\beta_{1t}, \beta_2$, a Stochastic First-order Oracle (SFO) $V(z; \xi)$, a sequence of batch sizes $\{M_k\}$.
**Output:** $z_t$ where $t$ is uniformly chosen from $\{0, 1, \ldots, N-1\}$.

1: **for** $k = 1, \ldots, N$ **do**
2:     (Gradient Evaluation 1) $g_{k-1} = \frac{1}{M_k} \sum_{i=1}^{M_k} V(z_{k-1}; \xi_{k-1}^i)$.
3:     (Momentum Update 1) $m_k = \beta_{1k} \hat{m}_{k-1} + (1 - \beta_{1k}) g_{k-1}$.
4:     (Velocity Update 1) $v_k = \max(\beta_2 \hat{v}_{k-1} + (1 - \beta_2) g_{k-1}^2, \hat{v}_{k-1})$, $H_k = \delta I + \text{Diag}(\sqrt{v_k})$.
5:     (Shadow Update) $\hat{z}_k = z_{k-1} + \eta \cdot H_k^{-1} m_k$.
6:     (Gradient Evaluation 2) $\hat{g}_k = \frac{1}{M_k} \sum_{i=1}^{M_k} V(\hat{z}_k; \xi_k^i)$.
7:     (Momentum Update 2) $\hat{m}_k = \beta_{1k} m_k + (1 - \beta_{1k}) \hat{g}_k$.
8:     (Velocity Update 2) $\hat{v}_k = \max(\beta_2 v_k + (1 - \beta_2) \hat{g}_k^2, v_k)$, $\hat{H}_k = \delta I + \text{Diag}(\sqrt{\hat{v}_k})$
9:     (Real Update) $z_k = z_{k-1} + \eta \cdot \hat{H}_k^{-1} \hat{m}_k$.
10: **end for**

---

Compared to the original Adam, we just add an extra-gradient technique and a taking-max process in velocity updates. It's worth mentioned that if we delete the maximizing operation in velocity update steps, then this algorithm degenerates to Extra-Gradient Adam, since the largest difference between Adam and AMSGrad is that the latter one guarantees that the velocity term is non-decreasing.

**Theorem 3.1** (Main Theorem 1). *For the AMSGrad-EG algorithm, given the objective function $\phi(x, y) : \mathbb{R}^{n_1 + n_2} \to \mathbb{R}$ and $V(z) = (-\nabla_x \phi, \nabla_y \phi)$ that satisfy Assumption 1 and Assumption 2, as well as the initial point $z_0 \in \mathcal{Z}$, the iteration number $N$, a sequence of batch sizes $\{M_k\}$ and a constant learning rate $\eta \leqslant \frac{\delta}{3L}$, then the output of the algorithm satisfies the following inequality:*

$$\mathbb{E}\|V(z)\|_2^2 \leqslant \frac{1}{N}\left[\frac{6dD^2(\delta+G)^2}{\eta^2} + \frac{12dG^2(\delta+G)^2}{\delta^2}\right] + \frac{150\sigma^2(\delta+G)}{N\delta}\sum_{t=1}^N \frac{1}{M_t}$$
$$+ \frac{48GD(\delta+G)}{N\eta}\sum_{t=1}^N \beta_{1t} + \frac{108\eta^2 G^2(\delta+G)}{N\delta}\sum_{t=1}^N \beta_{1t}^2.$$

Here, we analyze the conclusion above on two sides: parameter choosing on $\beta_{1k}$ and on $M_k$.

**Discussion** Here, we give some discussions on Theorem 3.1 and compare it with existing results. (1) There are two practical ways to choose the parameter sequence $\{\beta_{1t}\}$: (1) $\beta_{1t} = \beta_1 \cdot \lambda^{t-1}$ where $\beta_1, \lambda \in (0,1)$ and (2) $\beta_{1t} = 1/t$. In both settings, $\sum_{t=1}^N \beta_{1t}^2 = \mathcal{O}(1)$ and $\sum_{t=1}^N \beta_{1t} = \widetilde{\mathcal{O}}(1)$. Therefore, we can conclude from Theorem 3.1 that: $\mathbb{E}\|V(z)\|_2^2 \leqslant \mathcal{O}(d/N) + \mathcal{O}(1/N) \cdot \sum_{t=1}^N 1/M_t$ holds after regarding $D, G, \delta, \eta$ as constants.
(2) When the batch sizes $M_k$ are constant, let $M_k = \Theta(1/\varepsilon^2)$. To guarantee $\mathbb{E}\|V(z)\|_2^2 \leqslant \varepsilon^2$, the total number of iterations should be $N = \mathcal{O}(d\varepsilon^{-2})$ and the total complexity is $\sum_{k=0}^N M_k = \mathcal{O}(d\varepsilon^{-4})$. When the batch sizes $M_k$ are increasing, let $M_k = k + 1$. To guarantee $\mathbb{E}\|V(z)\|_2^2 \leqslant \varepsilon^2$, the total number of iterations should be $N = \widetilde{\mathcal{O}}(d\varepsilon^{-2})$ and the total complexity is $\sum_{k=0}^N M_k = \widetilde{\mathcal{O}}(d^2\varepsilon^{-4})$. Obviously, using constant batch sizes obtains a better total complexity.
(3) In the deterministic setting, the first-order oracle directly outputs the accurate gradient $V(z;\xi) = V(z)$, which means $\sigma = 0$. Theorem 3.1 leads to $\mathbb{E}\|V(z)\|_2^2 \leqslant \mathcal{O}(d/N)$. To guarantee $\mathbb{E}\|V(z)\|_2^2 \leqslant \varepsilon^2$, the total number of iterations should be $N = \mathcal{O}(d\varepsilon^{-2})$.
(4) In the AMSGrad-EG algorithm, the momentum term is a technical difficulty on the convergence proof. Proofs in the past works always use the MVI condition or convex condition like $\langle V(z_k), z_k - z^*\rangle \leqslant 0 \Rightarrow \langle g_k, z_k - z^*\rangle \lessgtr 0$ to control the gradient norms. However, if we replace $g_k$ with the momentum term $m_k$, the inequality above will no longer hold, and then we have to find another way to control the upper bound of gradient norms. It's also worth mentioned that our proof can't be extended to Optimistic Adam (OAdam) since we need to guarantee that $H_1 \preceq H_2 \preceq \ldots$ in our proof. Actually, Adam may not even converge in convex case (Reddi et al., 2018).
(5) Comparison with OAdaGrad: (Liu et al., 2020) proposes the Optimistic AdaGrad (OAdaGrad) algorithm and gives a convergence analysis on under Assumption 1, 2 and Bounded Cumulative Gradient Assumption (which assumes the existence of a constant $0 \leqslant \delta \leqslant 1/2$ such that the cumulative gradients are bounded as $\|\hat{g}_{1:k,i}\|_2 \leqslant \delta k^\alpha$ for all $k$). Under these assumptions, they conclude that:
$$\frac{1}{N}\sum_{k=1}^N \mathbb{E}\|V(z_k)\|_{H_{k-1}^{-1}}^2 \leqslant \mathcal{O}(1/N^{1-\alpha}).$$

On one hand, notice that $\frac{1}{N}\sum_{k=1}^N \mathbb{E}\|V(z_k)\|_{H_{k-1}^{-1}}^2$ is the average of the norms of $V(z_k)$. However, the norm keeps changing. Since $H_{k-1}^{-1}$ keeps decreasing and may limit to 0 as $k \to \infty$, its unclear what is the real convergence rate in terms of the size of the gradient. It would be more convincing if we can upper bound the average of constant norms like $\frac{1}{N}\sum_{k=1}^N \mathbb{E}\|V(z_k)\|_2^2$. On the other hand, the Bounded Cumulative Assumption though widely used in related papers (Zhou et al., 2018; Reddi et al., 2018; Duchi et al., 2011), is actually a very strong assumption: Under this assumption, it holds that $\|\hat{g}_{1:k,i}\|_2 \leqslant \delta k^\alpha$, which naturally leads to:
$$\frac{1}{N}\sum_{k=1}^N \mathbb{E}\|V(z_k)\|_2^2 \leqslant \frac{1}{N}\sum_{k=1}^N \mathbb{E}\|\hat{g}_k\|_2^2 \leqslant \mathcal{O}\left(\frac{d}{N^{1-2\alpha}}\right),$$

which causes circularity on the argument. In this paper, we overcome these two shortcomings.

**Standard MVI and one-sided MVI conditions** From Table 1, we can see that many related convergence proofs rely on assuming the MVI condition of $-V(z)$, which means $\langle V(z), z - z^*\rangle \leqslant 0 \; \forall z \in \mathcal{Z}$. Although MVI condition is theoretically known to be true in many standard supervised deep learning settings (Li & Yuan, 2017; Kleinberg et al., 2018; Allen-Zhu & Li, 2020; Allen-Zhu & Li, 2020; Li et al., 2018; 2020; Allen-Zhu & Li, 2020; 2019). However, this is a rather unrealistic assumption for GANs: In some practical scenarios such as DCGAN (Radford et al., 2015), it is unclear whether the training objective can satisfy the MVI condition: While the generator might have a consistent gradient direction towards the optimal generator (which is the one that generates the target distribution), it is very unlikely that there is a "optimal discriminator" where the discriminator's gradient is pointing to through the course of the training. Indeed, different generator

should in principle requires different discriminator to discriminate it from the target distribution, which precludes the MVI condition to hold on $y$.

Also, in practical scenarios like GAN, we only care about the min-variable $x$ (which refers to the generator of GAN), and the optimally of $y$ is not needed. Therefore, in the following part, we propose a weaker version of MVI condition, which is the one-sided MVI condition. Recall that $z = (x, y), z^* = (x^*, y^*)$ where $x \in \mathcal{X}, y \in \mathcal{Y}, \mathcal{Z} = \mathcal{X} \times \mathcal{Y}$, and $V(z) = (-\nabla_x \phi(z), \nabla_y \phi(z)) := (V_x(z), V_y(z))$. Then, the one-sided MVI condition implies that $\langle V_x(z), x - x^* \rangle \leqslant 0 \quad \forall z \in \mathcal{Z}$, which means for any $y \in \mathcal{Y}$, the $x$-part of function $V$, $-V(\cdot, y)$ satisfies the MVI condition. Now we empirically verify that one-sided MVI is more likely to hold in practice in some simple applications of GANs. For $z, z^* \in \mathcal{Z}$:

$$\langle -V(z), z - z^* \rangle = \langle -V_x(z), x - x^* \rangle + \langle -V_y(z), y - y^* \rangle,$$

where $z = (x, y), z^* = (x^*, y^*)$. We call the three terms above as total MVI, $x$-sided MVI and $y$-sided MVI respectively. Assumption 2 requires total MVI to be non-negative, and Assumption 3 requires $x$-sided MVI to be non-negative. After training Wasserstein GAN on the MNIST/Fashion MNIST dataset with AMSGrad-EG optimizer, we denote $z_k := (x_k, y_k)$ as the value of $z$ at the $k$-th iteration, and $z^* := (x^*, y^*)$ as the value of $z$ at the last iteration. In the following Figure 2, we plot the total MVI values $\langle -V(z_k), z_k - z^* \rangle$, $x$-sided MVI values $\langle -V_x(z_k), x_k - x^* \rangle$, and $y$-sided MVI values $\langle -V_y(z_k), y_k - y^* \rangle$ along the training trajectory. We can see that $x$-sided MVI stays positive while the total MVI does not, which means the one-sided MVI condition proposed in Assumption 3 is more realistic than the original MVI condition in Assumption 2. Under the one-sided MVI condition,

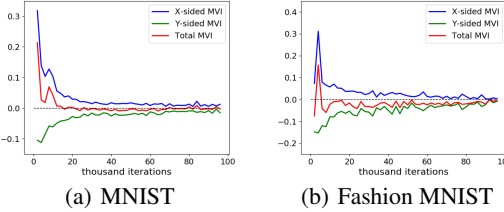

(a) MNIST       (b) Fashion MNIST

Figure 2: This figure shows the MVI values along the training trajectory. As we can see, the blue curve stays above $x$ axis in both experiments while the red curve does not. Since we use non-linear activations in the network architecture, this result is exciting. It's safe to say that the one-sided MVI condition proposed by us fits the reality since the $x$-sided MVI keeps positive.

we prove in our next theorem that, the conclusion of Theorem 3.1 still holds once we slightly modify AMSGrad-EG to Extra Gradient AMSGrad with Dual Rate Decay (AMSGrad-EG-DRD). To the best of our knowledge, this is a convergence guarantee of an adaptive min-max algorithm with the weakest assumption ever needed. In the next section, we introduce the AMSGrad-EG-DRD algorithm and its convergence property.

### 3.3 Extra Gradient AMSGrad with Dual Rate Decay

Now, we write down the one-sided MVI condition introduced above in Assumption 3, which is the weakest assumption ever needed to obtain a convergence guarantee in min-max optimization.

**Assumption 3** (One-sided MVI condition). *The one-sided MVI of $-V(z)$ has a solution, which means there exists a $z^* = (x^*, y^*) \in \mathcal{Z}$, such that:*

$$\langle -V_x(z), x - x^* \rangle \geqslant 0 \ holds \ for \ \forall z = (x, y) \in \mathcal{Z}.$$

After slightly modifying AMSGrad-EG with a $\mathcal{O}(1/\sqrt{k})$ dual rate decay, we get Extra Gradient AMSGrad with dual rate decay (AMSGrad-EG-DRD). Its pseudo-algorithm is placed in the appendix. We propose its convergence property as follows:

**Theorem 3.2** (Main Theorem 2). *For the AMSGrad with Extra-Gradient and Dual Rate Decay (AMSGrad-EG-DRD) algorithm, given the objective function $\phi(x, y) : \mathbb{R}^{n_1+n_2} \to \mathbb{R}$ and $V(z) = (-\nabla_x \phi, \nabla_y \phi) := (V_x(z), V_y(z))$ that satisfy Assumption 1 and Assumption 3, as well as the initial point $z_0 \in \mathcal{Z}$, the iteration number $N$, a sequence of batch sizes $\{M_k\}$ and a constant learning rate $\eta \leqslant \frac{\delta}{3L}$, then the output of the algorithm satisfies the following inequality:*

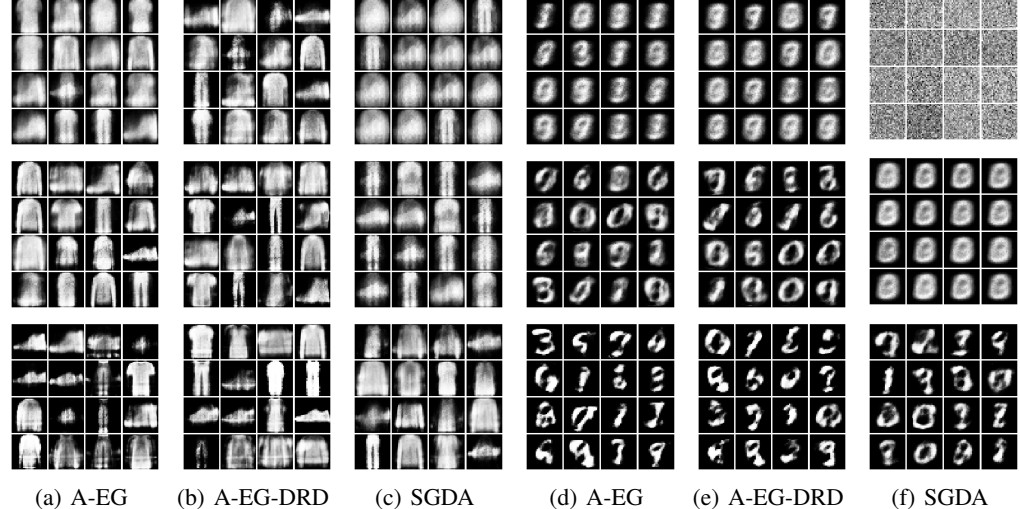

| (a) A-EG | (b) A-EG-DRD | (c) SGDA | (d) A-EG | (e) A-EG-DRD | (f) SGDA |

Figure 3: Generated MNIST and Fashion-MNIST figures by the three algorithms after 10k, 20k, 50k iterations. A-EG and A-EG-DRD stand for AMSGrad-EG and AMSGrad-EG-DRD.

$$\mathbb{E}\|V_x(z)\|_2^2 \leqslant \frac{(15 + 3\log N)dG^2(\delta + G)^2}{N\delta^2} + \frac{6dD^2(\delta + G)^2}{N\eta^2} + \frac{150\sigma^2(\delta + G)}{N\delta}\sum_{t=1}^{N}\frac{1}{M_t}$$

$$+ \frac{48GD(\delta + G)}{N\eta}\sum_{t=1}^{N}\beta_{1t} + \frac{108\eta^2G^2(\delta + G)}{N\delta}\sum_{t=1}^{N}\beta_{1t}^2.$$

Similar to Theorem 3.1, in the deterministic setting where $\sigma = 0$, the total complexity is $\widetilde{\mathcal{O}}(d\varepsilon^{-2})$. In the stochastic setting, we have: $\mathbb{E}\|V_x(z)\|_2^2 \leqslant \widetilde{\mathcal{O}}(d/N) + \mathcal{O}(1/N) \cdot \sum_{t=1}^{N} 1/M_t$. When we use constant batch sizes $M_k = \Theta(1/\varepsilon^2)$, iteration number $N$ should be $\widetilde{\mathcal{O}}(d\varepsilon^{-2})$ in order to guarantee that $\mathbb{E}\|V_x(z)\|_2^2 \leqslant \varepsilon^2$. So the total complexity should be $\widetilde{\mathcal{O}}(d\varepsilon^{-4})$.

## 4 EXPERIMENTAL RESULTS

In this section, we use experiments to verify the effectiveness of AMSGrad-EG and AMSGrad-EG-DRD algorithms by applying Wasserstein GAN (Arjovsky et al., 2017) on the MNIST (LeCun et al., 1998) and Fashion-MNIST (Xiao et al., 2017) datasets in our experiments. More experiments will be shown in the appendix. The architectures of discriminator and generator are set to be MLP. The layer widths of generator MLP are 100, 128, 784 and the layer widths of discriminator MLP are 784, 128, 1. We set batch sizes as 64, learning rate as $1e$-4 and we compare AMSGrad-EG, AMSGrad-EG-DRD and SGDA by printing their generated figures after 10k, 20k, 50k iterations in the following Figure 3. We use the Tensorflow framework (Abadi et al., 2016) to complete our experiments. As a result, unlike the non-adaptive SGDA algorithm, the two algorithms proposed by us perform better than the non-adaptive SGDA and their generated figures are realistic, which shows their effectiveness.

## 5 DISCUSSION AND FUTURE WORKS

This work fills up the blank in the theory of non-convex non-concave min-max optimization as well as GAN's training. We bridge the gap between theory and practice and provide the theoretical guarantee of the one-sided convergence of Adam under one-sided MVI condition, which perfectly matches the empirical observation. To the best of our knowledge, it is the very first proof for the convergence of Adam-type algorithms in non-convex non-concave min-max optimization. Future follow-up works can go further on the following two directions: (a) Figure out which part of Adam-type algorithms play an important role on the outstanding performance: automatic tuning of learning rate or local geometry adaptivity. (b) With both the discriminator and generator of GANs to be overparameterized 2-layer ReLU networks, it would be an influential work to figure out the convergence property, the converging limit and the training trajectory of min-max optimization under multiple optimizers like Adam and SGDA so that we can get some intuition on their differences.

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

## A   PROOF FOR THE CONVERGENCE OF AMSGRAD-EG

We recall that in the $t$-th iteration of Extra-Gradient AMSGrad, our update is as follows:

$$
\begin{aligned}
&m_t = \beta_{1t}\hat{m}_{t-1} + (1-\beta_{1t})g_{t-1}, \; v_t = \max(\beta_2\hat{v}_{t-1} + (1-\beta_2)g_{t-1}^2, \hat{v}_{t-1}) \\
&H_t = \delta I + \mathrm{Diag}(\sqrt{v_t}), \; \hat{z}_t = z_{t-1} + \eta \cdot H_t^{-1}m_t \\
&\hat{m}_t = \beta_{1t}m_t + (1-\beta_{1t})\hat{g}_t, \; \hat{v}_t = \max(\beta_2 v_t + (1-\beta_2)\hat{g}_t^2, v_t) \\
&\hat{H}_t = \delta I + \mathrm{Diag}(\sqrt{\hat{v}_t}), \; z_t = z_{t-1} + \eta \cdot \hat{H}_t^{-1}\hat{m}_t.
\end{aligned}
\tag{1}
$$

Now we begin to prove Theorem 3.1 (Main Theorem 1). Before that, we prove that as the weighted sum of stochastic gradient, the momentum terms $m_t$, $\hat{m}_t$ are also contained in $l_2$ ball with radius $G$.

**Lemma A.1.** *There exist upper bounds for both velocity terms $v_t$, $\hat{v}_t$ and momentum terms $m_t$, $\hat{m}_t$:*
*(1) For $\forall t \in \mathbb{N}$, almost surely, the momentum terms $\|m_t\|_2 \leqslant G$, $\|\hat{m}_t\|_2 \leqslant G$.*
*(2) For $\forall t \in \mathbb{N}$, almost surely, the velocity terms $|v_{t,i}| \leqslant G^2$, $|\hat{v}_{t,i}| \leqslant G^2$ hold for $\forall i \in [d]$.*

*Proof of Lemma A.1.* Actually, this lemma can be simply proved by using the method of induction. Since we've assumed that $\|V(z)\|_2 \leqslant G$ almost surely holds for $z \in \mathcal{Z} = \mathbb{R}^d$, so $\|g_t\|_2 \leqslant G$ and $\|\hat{g}_t\|_2 \leqslant G$ almost surely holds. Therefore, by knowing that $m_0 = \hat{m}_0 = 0 \Rightarrow 0 = \|m_0\|_2 = \|\hat{m}_0\| \leqslant G$ and once $\|m_{k-1}\|_2 \leqslant G$, $\|\hat{m}_{k-1}\|_2 \leqslant G$, we have:

$$\|m_k\|_2 = \|\beta_{1k}\hat{m}_{k-1} + (1-\beta_{1k})g_{k-1}\|_2 \leqslant \beta_{1k}\|\hat{m}_{k-1}\|_2 + (1-\beta_{1k})\|g_{k-1}\|_2 \leqslant (1-\beta_{1k})G + \beta_{1k}G = G$$
$$\|\hat{m}_k\|_2 = \|\beta_{1k}m_k + (1-\beta_{1k})\hat{g}_k\|_2 \leqslant \beta_{1k}\|m_k\|_2 + (1-\beta_{1k})\|\hat{g}_k\|_2 \leqslant (1-\beta_{1k})G + \beta_{1k}G = G.$$

Similarly, the upper bound for velocity terms can also be easily proved.

$\square$

**Lemma A.2.**

$$\|z_t - z^*\|_{\hat{H}_t}^2 \leqslant \|z_{t-1} - z^*\|_{\hat{H}_t}^2 - \|z_{t-1} - \hat{z}_t\|_{\hat{H}_t}^2 + \|\hat{z}_t - z_t\|_{\hat{H}_t}^2 + 2\langle \eta \cdot \hat{\varepsilon}_t, \hat{z}_t - z^* \rangle + 8\eta\beta_{1t}GD.$$

*Here, $\hat{\varepsilon}_t = \hat{g}_t - V(\hat{z}_t) = \frac{1}{M_t}\sum_{i=1}^{M_t}V(\hat{z}_t; \hat{\xi}_i) - V(\hat{z}_t)$.*

*Proof of Lemma A.2.* According to the update rules:

$$
\begin{aligned}
\|z_t - z^*\|_{\hat{H}_t}^2 &= \|z_{t-1} + \eta \cdot \hat{H}_t^{-1}\hat{m}_t - z^*\|_{\hat{H}_t}^2 \\
&= \|z_{t-1} + \eta \cdot \hat{H}_t^{-1}\hat{m}_t - z^*\|_{\hat{H}_t}^2 - \|z_{t-1} + \eta \cdot \hat{H}_t^{-1}\hat{m}_t - z_t\|_{\hat{H}_t}^2 \\
&= \|z_{t-1} - z^*\|_{\hat{H}_t}^2 - \|z_{t-1} - z_t\|_{\hat{H}_t}^2 + 2\langle \eta \cdot \hat{m}_t, z_t - z^* \rangle \\
&= \|z_{t-1} - z^*\|_{\hat{H}_t}^2 - \|z_{t-1} - \hat{z}_t + \hat{z}_t - z_t\|_{\hat{H}_t}^2 + 2\langle \eta \cdot \hat{m}_t, z_t - \hat{z}_t \rangle + 2\langle \eta \cdot \hat{m}_t, \hat{z}_t - z^* \rangle \\
&= \|z_{t-1} - z^*\|_{\hat{H}_t}^2 - \|z_{t-1} - \hat{z}_t\|_{\hat{H}_t}^2 - \|\hat{z}_t - z_t\|_{\hat{H}_t}^2 - 2\langle \hat{H}_t(z_{t-1} - \hat{z}_t), \hat{z}_t - z_t \rangle \\
&\quad + 2\langle \eta \cdot \hat{m}_t, z_t - \hat{z}_t \rangle + 2\langle \eta \cdot \hat{m}_t, \hat{z}_t - z^* \rangle \\
&= \|z_{t-1} - z^*\|_{\hat{H}_t}^2 - \|z_{t-1} - \hat{z}_t\|_{\hat{H}_t}^2 - \|\hat{z}_t - z_t\|_{\hat{H}_t}^2 + 2\langle \eta \cdot \hat{m}_t, \hat{z}_t - z^* \rangle \\
&\quad + 2\langle z_t - \hat{z}_t, \hat{H}_t(z_{t-1} - \hat{z}_t + \eta \cdot \hat{H}_t^{-1}\hat{m}_t) \rangle \\
&= \|z_{t-1} - z^*\|_{\hat{H}_t}^2 - \|z_{t-1} - \hat{z}_t\|_{\hat{H}_t}^2 - \|\hat{z}_t - z_t\|_{\hat{H}_t}^2 + 2\langle \eta \cdot \hat{m}_t, \hat{z}_t - z^* \rangle \\
&\quad + 2\langle z_t - \hat{z}_t, \hat{H}_t(z_t - \hat{z}_t) \rangle \\
&= \|z_{t-1} - z^*\|_{\hat{H}_t}^2 - \|z_{t-1} - \hat{z}_t\|_{\hat{H}_t}^2 + \|\hat{z}_t - z_t\|_{\hat{H}_t}^2 + 2\langle \eta \cdot \hat{m}_t, \hat{z}_t - z^* \rangle
\end{aligned}
$$

Notice that $\hat{g}_t = V(\hat{z}_t) + \hat{\varepsilon}_t$. Since $z^*$ is a solution of MVI which means $\langle V(z), z - z^* \rangle \leqslant 0$ holds for $\forall z \in \mathcal{Z}$, so $\langle V(\hat{z}_t), \hat{z}_t - z^* \rangle \leqslant 0$. Therefore:

$$
\begin{aligned}
\langle \eta \cdot \hat{m}_t, \hat{z}_t - z^* \rangle &= \langle \eta \cdot (\beta_{1t}m_t + (1-\beta_{1t})\hat{g}_t), \hat{z}_t - z^* \rangle \leqslant \langle \eta \cdot \hat{g}_t, \hat{z}_t - z^* \rangle + \langle \eta \cdot \beta_{1t}(m_t - \hat{g}_t), \hat{z}_t - z^* \rangle \\
&\leqslant \langle \eta \cdot (V(\hat{z}_t) + \hat{\varepsilon}_t), \hat{z}_t - z^* \rangle + \eta\beta_{1t} \cdot \|m_t - \hat{g}_t\|_2 \cdot \|\hat{z}_t - z^*\|_2 \\
&\overset{(a)}{\leqslant} \langle \eta\hat{\varepsilon}_t, \hat{z}_t - z^* \rangle + 4\eta\beta_{1t}GD.
\end{aligned}
$$

Here, $(a)$ holds because $\|m_t - \hat{g}_t\|_2 \leqslant \|m_t\|_2 + \|\hat{g}_t\|_2 \leqslant 2G$, $\|\hat{z}_t - z^*\|_2 \leqslant 2D$ and by using MVI property, $\langle V(\hat{z}_t), \hat{z}_t - z^* \rangle \leqslant 0$.

Combine it with the inequality above, we obtain that:

$$\|z_t - z^*\|_{\hat{H}_t}^2 \leqslant \|z_{t-1} - z^*\|_{\hat{H}_t}^2 - \|z_{t-1} - \hat{z}_t\|_{\hat{H}_t}^2 + \|\hat{z}_t - z_t\|_{\hat{H}_t}^2 + 2\langle \eta \cdot \hat{\varepsilon}_t, \hat{z}_t - z^* \rangle + 8\eta\beta_{1t}GD.$$

which comes to our conclusion. $\qquad\square$

In the lemma above, $\langle \eta \cdot \hat{\varepsilon}_t, \hat{z}_t - z^* \rangle$ has zero mean. So it can be ignoring when taking expectation. Next, we upper bound the $\|\hat{z}_t - z_t\|_{\hat{H}_t}^2$ term.

**Lemma A.3.**

$$\|\hat{z}_t - z_t\|_{\hat{H}_t}^2 \leqslant 2\eta^2 \cdot \|(\hat{H}_t^{-1} - H_t^{-1})m_t\|_{\hat{H}_t}^2 + \frac{16\eta^2 G^2 \beta_{1t}^2}{\delta} + \frac{12\eta^2 L^2}{\delta^2} \cdot \|\hat{z}_t - z_{t-1}\|_{\hat{H}_t}^2$$

$$+ 12\eta^2 \cdot \left( \|\hat{\varepsilon}_t\|_{\hat{H}_t^{-1}}^2 + \|\varepsilon_{t-1}\|_{H_t^{-1}}^2 \right).$$

*Here, $\varepsilon_{t-1} = g_{t-1} - V(z_{t-1})$ and $\hat{\varepsilon}_t = \hat{g}_t - V(\hat{z}_t)$.*

*Proof of Lemma A.3.* According to the update rules (1), we know that $z_t - \hat{z}_t = \eta \cdot (\hat{H}_t^{-1}\hat{m}_t - H_t^{-1}m_t)$. Therefore, we upper bound the term $\|\hat{z}_t - z_t\|_{\hat{H}_t}^2$ as follows:

$$\|\hat{z}_t - z_t\|_{\hat{H}_t}^2 = \eta^2 \cdot \|\hat{H}_t^{-1}\hat{m}_t - H_t^{-1}m_t\|_{\hat{H}_t}^2 = \eta^2 \cdot \|\hat{H}_t^{-1}(\hat{m}_t - m_t) + (\hat{H}_t^{-1} - H_t^{-1})m_t\|_{\hat{H}_t}^2$$

$$\leqslant 2\eta^2 \cdot \left( \|\hat{m}_t - m_t\|_{\hat{H}_t^{-1}}^2 + \|(\hat{H}_t^{-1} - H_t^{-1})m_t\|_{\hat{H}_t}^2 \right)$$

$$\overset{(a)}{=} 2\eta^2 \cdot \|(\hat{H}_t^{-1} - H_t^{-1})m_t\|_{\hat{H}_t}^2 + 2\eta^2 \cdot \|\beta_{1t}(m_t - \hat{m}_{t-1}) + (1 - \beta_{1t})(\hat{g}_t - g_{t-1})\|_{\hat{H}_t^{-1}}^2$$

$$\overset{(b)}{\leqslant} 2\eta^2 \cdot \|(\hat{H}_t^{-1} - H_t^{-1})m_t\|_{\hat{H}_t}^2 + 4\eta^2 \left( \beta_{1t}^2 \|m_t - \hat{m}_{t-1}\|_{\hat{H}_t^{-1}}^2 + (1 - \beta_{1t})^2 \|\hat{g}_t - g_{t-1}\|_{\hat{H}_t^{-1}}^2 \right)$$

$$\overset{(c)}{\leqslant} 2\eta^2 \cdot \|(\hat{H}_t^{-1} - H_t^{-1})m_t\|_{\hat{H}_t}^2 + \frac{16\eta^2 \beta_{1t}^2 G^2}{\delta} + 4\eta^2 \|\hat{g}_t - g_{t-1}\|_{\hat{H}_t^{-1}}^2$$

$$\overset{(d)}{=} 2\eta^2 \cdot \|(\hat{H}_t^{-1} - H_t^{-1})m_t\|_{\hat{H}_t}^2 + \frac{16\eta^2 G^2 \beta_{1t}^2}{\delta} + 4\eta^2 \|V(\hat{z}_t) - V(z_{t-1}) + \hat{\varepsilon}_t - \varepsilon_{t-1}\|_{\hat{H}_t^{-1}}^2$$

$$\overset{(e)}{\leqslant} 2\eta^2 \cdot \|(\hat{H}_t^{-1} - H_t^{-1})m_t\|_{\hat{H}_t}^2 + \frac{16\eta^2 G^2 \beta_{1t}^2}{\delta} + 12\eta^2 \cdot \|V(\hat{z}_t) - V(z_{t-1})\|_{\hat{H}_t^{-1}}^2$$

$$+ 12\eta^2 \cdot \left( \|\hat{\varepsilon}_t\|_{\hat{H}_t^{-1}}^2 + \|\varepsilon_{t-1}\|_{\hat{H}_t^{-1}}^2 \right)$$

$$\overset{(f)}{\leqslant} 2\eta^2 \cdot \|(\hat{H}_t^{-1} - H_t^{-1})m_t\|_{\hat{H}_t}^2 + \frac{16\eta^2 G^2 \beta_{1t}^2}{\delta} + \frac{12\eta^2 L^2}{\delta^2} \cdot \|\hat{z}_t - z_{t-1}\|_{\hat{H}_t}^2$$

$$+ 12\eta^2 \cdot \left( \|\hat{\varepsilon}_t\|_{\hat{H}_t^{-1}}^2 + \|\varepsilon_{t-1}\|_{H_t^{-1}}^2 \right)$$

Here, $(a)$ holds because $m_t = \beta_{1t}\hat{m}_{t-1} + (1 - \beta_{1t})g_{t-1}$, $\hat{m}_t = \beta_{1t}m_t + (1 - \beta_{1t})\hat{g}_t$, and then:

$$\hat{m}_t - m_t = \beta_{1t}(m_t - \hat{m}_{t-1}) + (1 - \beta_{1t})(\hat{g}_t - g_{t-1}).$$

$(b)$ holds because $\|a + b\|_C^2 \leqslant (\|a\|_C + \|b\|_C)^2 \leqslant 2(\|a\|_C^2 + \|b\|_C^2)$ and $(e)$ holds because of the similar reason: $\|x + y + z\|_C^2 \leqslant (\|x\|_C + \|y\|_C + \|z\|_C)^2 \leqslant 3(\|x\|_C^2 + \|y\|_C^2 + \|z\|_C^2)$. $(c)$ holds because $(1 - \beta_{1t})^2 < 1$ and

$$\|m_t - \hat{m}_{t-1}\|_{\hat{H}_t^{-1}}^2 \leqslant \frac{\|m_t - \hat{m}_{t-1}\|^2}{\delta} \leqslant \frac{4G^2}{\delta}.$$

$(d)$ holds because $\hat{g}_t = V(\hat{z}_t) + \hat{\varepsilon}_t$ and $g_{t-1} = V(z_{t-1}) + \varepsilon_{t-1}$. $(f)$ holds because of the following fact:

$$\delta I = H_0 \preceq \hat{H}_0 \preceq H_1 \preceq \hat{H}_1 \preceq \ldots \preceq \hat{H}_N \preceq \ldots,$$

or equivalently:

$$\frac{1}{\delta}I \succeq H_0^{-1} \succeq \hat{H}_0^{-1} \succeq H_1^{-1} \succeq \hat{H}_1^{-1} \succeq \ldots \succeq \hat{H}_N^{-1} \succeq \ldots,$$

which leads to $\|\varepsilon_{t-1}\|^2_{\hat{H}_t^{-1}} \leqslant \|\varepsilon_{t-1}\|^2_{H_t^{-1}}$. Also, $V(\cdot)$ is $L$-Lipschitz continuous and $\delta I \preceq \hat{H}_t$, so:

$$\|V(\hat{z}_t) - V(z_{t-1})\|^2_{\hat{H}_t^{-1}} \leqslant \frac{L^2}{\delta^2}\|\hat{z}_t - z_{t-1}\|^2_{\hat{H}_t}.$$

$\square$

Since our learning rate $\eta \leqslant \frac{\delta}{5L}$, we have $\frac{12\eta^2 L^2}{\delta^2} \leqslant \frac{1}{2}$. Now, we can combine Lemma A.2 and Lemma A.3:

$$\|z_t - z^*\|^2_{\hat{H}_t} \leqslant \|z_{t-1} - z^*\|^2_{\hat{H}_t} - \|z_{t-1} - \hat{z}_t\|^2_{\hat{H}_t} + 2\langle\eta \cdot \hat{\varepsilon}_t, \hat{z}_t - z^*\rangle + 8\eta\beta_{1t}GD$$
$$+ 2\eta^2 \cdot \|(\hat{H}_t^{-1} - H_t^{-1})m_t\|^2_{\hat{H}_t} + \frac{16\eta^2 G^2\beta_{1t}^2}{\delta} + \frac{12\eta^2 L^2}{\delta^2} \cdot \|\hat{z}_t - z_{t-1}\|^2_{\hat{H}_t}$$
$$+ 12\eta^2 \cdot \left(\|\hat{\varepsilon}_t\|^2_{\hat{H}_t^{-1}} + \|\varepsilon_{t-1}\|^2_{H_t^{-1}}\right),$$

Since $\frac{12\eta^2 L^2}{\delta^2} \leqslant \frac{1}{2}$, therefore:

$$\|z_t - z^*\|^2_{\hat{H}_t} \leqslant \|z_{t-1} - z^*\|^2_{\hat{H}_t} - \frac{1}{2}\|z_{t-1} - \hat{z}_t\|^2_{\hat{H}_t} + 2\langle\eta \cdot \hat{\varepsilon}_t, \hat{z}_t - z^*\rangle + 8\eta\beta_{1t}GD + \frac{16\eta^2 G^2\beta_{1t}^2}{\delta}$$
$$+ 2\eta^2 \cdot \|(\hat{H}_t^{-1} - H_t^{-1})m_t\|^2_{\hat{H}_t} + 12\eta^2 \cdot \left(\|\hat{\varepsilon}_t\|^2_{\hat{H}_t^{-1}} + \|\varepsilon_{t-1}\|^2_{H_t^{-1}}\right).$$

After taking expectation and summation over $t = 1, 2, \ldots, N$, we obtain that:

$$\frac{1}{2}\sum_{t=1}^N \mathbb{E}\|z_{t-1} - \hat{z}_t\|^2_{\hat{H}_t} \leqslant \sum_{t=1}^N \mathbb{E}\left[\|z_{t-1} - z^*\|^2_{\hat{H}_t} - \|z_t - z^*\|^2_{\hat{H}_t}\right] + \sum_{t=1}^N \left[8\eta\beta_{1t}GD + \frac{16\eta^2 G^2\beta_{1t}^2}{\delta}\right]$$
$$+ 2\eta^2 \cdot \sum_{t=1}^N \mathbb{E}\|(\hat{H}_t^{-1} - H_t^{-1})m_t\|^2_{\hat{H}_t} + 12\eta^2 \cdot \sum_{t=1}^N \mathbb{E}\left[\|\hat{\varepsilon}_t\|^2_{\hat{H}_t^{-1}} + \|\varepsilon_{t-1}\|^2_{H_t^{-1}}\right].$$

$$(2)$$

Since $\|z_{t-1} - \hat{z}_t\|^2_{\hat{H}_t} = \|\eta \cdot H_t^{-1}m_t\|^2_{\hat{H}_t}$ and $V(z_{t-1}) = g_{t-1} - \varepsilon_{t-1} = m_t - \beta_{1t}(\hat{m}_{t-1} - g_{t-1}) - \varepsilon_{t-1}$, we know that:

$$\|V(z_{t-1})\|^2_{H_t^{-1}} = \|m_t - \beta_{1t}(\hat{m}_{t-1} - g_{t-1}) - \varepsilon_{t-1}\|^2_{H_t^{-1}}$$
$$\leqslant 3\left(\|m_t\|^2_{H_t^{-1}} + \|\beta_{1t}(\hat{m}_{t-1} - g_{t-1})\|^2_{H_t^{-1}} + \|\varepsilon_{t-1}\|^2_{H_t^{-1}}\right)$$
$$= 3\left(\|\beta_{1t}(\hat{m}_{t-1} - g_{t-1})\|^2_{H_t^{-1}} + \|\varepsilon_{t-1}\|^2_{H_t^{-1}}\right) + 3 \cdot \|H_t^{-1}m_t\|^2_{\hat{H}_t} \qquad (3)$$
$$\leqslant \frac{12G^2\beta_{1t}^2}{\delta} + 3\|\varepsilon_{t-1}\|^2_{H_t^{-1}} + \frac{3}{\eta^2}\|z_{t-1} - \hat{z}_t\|^2_{\hat{H}_t}$$

After combining Equation (2) and Equation (3), it holds that:

$$\sum_{t=1}^N \mathbb{E}\|V(z_{t-1})\|^2_{H_t^{-1}} \leqslant \frac{6}{\eta^2}\sum_{t=1}^N \mathbb{E}\left[\|z_{t-1} - z^*\|^2_{\hat{H}_t} - \|z_t - z^*\|^2_{\hat{H}_t}\right] + \sum_{t=1}^N \left[\frac{48\beta_{1t}GD}{\eta} + \frac{108\eta^2 G^2\beta_{1t}^2}{\delta}\right]$$
$$+ 12\sum_{t=1}^N \mathbb{E}\|(\hat{H}_t^{-1} - H_t^{-1})m_t\|^2_{\hat{H}_t} + 75\sum_{t=1}^N \mathbb{E}\left[\|\hat{\varepsilon}_t\|^2_{\hat{H}_t^{-1}} + \|\varepsilon_{t-1}\|^2_{H_t^{-1}}\right].$$

$$(4)$$

In the following steps, we will upper bound the four terms above on the right side one by one, and we start from the first term.

**Lemma A.4.**

$$\sum_{t=1}^N \left[\|z_{t-1} - z^*\|^2_{\hat{H}_t} - \|z_t - z^*\|^2_{\hat{H}_t}\right] \leqslant dD^2 \cdot (\delta + G),$$

*which is a constant.*

*Proof of Lemma A.4.* Notice that

$$\sum_{t=1}^{N} \left[ \|z_{t-1} - z^*\|_{\hat{H}_t}^2 - \|z_t - z^*\|_{\hat{H}_t}^2 \right] \leqslant \|z_0 - z^*\|_{\hat{H}_0}^2 + \sum_{t=1}^{N-1} \left( \|z_t - z^*\|_{\hat{H}_{t+1}}^2 - \|z_t - z^*\|_{\hat{H}_t}^2 \right)$$

$$= \|z_0 - z^*\|_{\hat{H}_0}^2 + \sum_{t=1}^{N-1} \left[ (z_t - z^*)^\top (\hat{H}_{t+1} - \hat{H}_t) \cdot (z_t - z^*) \right]$$

$$\leqslant D^2 \cdot \mathrm{tr}(\hat{H}_0) + \sum_{t=1}^{N-1} D^2 \cdot (\mathrm{tr}(\hat{H}_{t+1}) - \mathrm{tr}(\hat{H}_t)) = D^2 \cdot \mathrm{tr}(\hat{H}_N) \leqslant dD^2 \cdot (\delta + G),$$

which comes to our conclusion. $\qquad\square$

For the second term, it's easy to see that:

$$\sum_{t=1}^{N} \left[ \frac{48\beta_{1t} GD}{\eta} + \frac{108\eta^2 G^2 \beta_{1t}^2}{\delta} \right] = \frac{48GD}{\eta} \sum_{t=1}^{N} \beta_{1t} + \frac{108\eta^2 G^2}{\delta} \sum_{t=1}^{N} \beta_{1t}^2, \tag{5}$$

Next, we analyze the third term.

**Lemma A.5.**

$$\sum_{t=1}^{N} \|(H_t^{-1} - \hat{H}_t^{-1})m_t\|_{\hat{H}_t}^2 \leqslant \frac{dG^2(\delta + G)}{\delta^2}.$$

*Proof of Lemma A.5.* For any $\delta \leqslant x < y$, we notice that:

$$y\left( \frac{1}{x} - \frac{1}{y} \right)^2 = \frac{(y-x)^2}{x^2 y} < \frac{y-x}{x^2} \leqslant \frac{y-x}{\delta^2}. \tag{6}$$

Therefore, we have:

$$\sum_{t=1}^{N} \|(H_t^{-1} - \hat{H}_t^{-1})m_t\|_{\hat{H}_t}^2 \leqslant \sum_{t=1}^{N} m_t^\top (H_t^{-1} - \hat{H}_t^{-1})\hat{H}_t (H_t^{-1} - \hat{H}_t^{-1})m_t$$

$$\leqslant \sum_{t=1}^{N} G^2 \cdot \mathrm{tr}\left( (H_t^{-1} - \hat{H}_t^{-1})\hat{H}_t (H_t^{-1} - \hat{H}_t^{-1}) \right)$$

$$\overset{(a)}{\leqslant} \sum_{t=1}^{N} \frac{G^2}{\delta^2} \cdot [\mathrm{tr}(\hat{H}_t) - \mathrm{tr}(H_t)]$$

$$\leqslant \frac{G^2}{\delta^2} \cdot \mathrm{tr}(\hat{H}_N) < \frac{G^2}{\delta^2} \cdot d(\delta + G) = \frac{dG^2(\delta + G)}{\delta^2}.$$

Here, $(a)$ holds because of Equation (6). $\qquad\square$

Finally, we come to the noise term $\sum_{t=1}^{N} \mathbb{E}\left[ \|\hat{\varepsilon}_t\|_{\hat{H}_t^{-1}}^2 + \|\varepsilon_{t-1}\|_{H_t^{-1}}^2 \right]$, which is closely related to our batch sizes $M_t$. Obviously, we have:

$$\mathbb{E}\|\hat{\varepsilon}_t\|_{\hat{H}_t^{-1}}^2 \leqslant \frac{1}{\delta} \cdot \mathbb{E}\|\hat{\varepsilon}_t\|_2^2 \leqslant \frac{1}{\delta} \cdot \frac{\sigma^2}{M_t}.$$

Similarly,

$$\mathbb{E}\|\varepsilon_{t-1}\|_{H_t^{-1}}^2 \leqslant \frac{1}{\delta} \cdot \frac{\sigma^2}{M_t}$$

Therefore, we can upper bound the expectation of the noise term as:

$$\sum_{t=1}^{N} \mathbb{E}\left[ \|\hat{\varepsilon}_t\|_{\hat{H}_t^{-1}}^2 + \|\varepsilon_{t-1}\|_{H_t^{-1}}^2 \right] \leqslant \frac{2\sigma^2}{\delta} \sum_{t=1}^{N} \frac{1}{M_t}. \tag{7}$$

Finally, after we combine Equation (4) with Equation (5), Equation (7) and Lemma A.4, Lemma A.5, we obtain that:

$$\sum_{t=1}^{N} \mathbb{E}\|V(z_{t-1})\|_{H_t^{-1}}^2 \leqslant \frac{6dD^2(\delta + G)}{\eta^2} + \frac{12dG^2(\delta + G)}{\delta^2} + \frac{150\sigma^2}{\delta} \sum_{t=1}^{N} \frac{1}{M_t}$$

$$+ \frac{48GD}{\eta} \sum_{t=1}^{N} \beta_{1t} + \frac{108\eta^2 G^2}{\delta} \sum_{t=1}^{N} \beta_{1t}^2.$$

Since for $\forall t$, $\|V(z_{t-1})\|_{H_t^{-1}} \geqslant \frac{1}{\delta + G}\|V(z_{t-1})\|_2$, therefore:

$$\sum_{t=1}^{N} \mathbb{E}\|V(z_{t-1})\|^2 \leqslant \frac{6dD^2(\delta + G)^2}{\eta^2} + \frac{12dG^2(\delta + G)^2}{\delta^2} + \frac{150\sigma^2(\delta + G)}{\delta} \sum_{t=1}^{N} \frac{1}{M_t}$$

$$+ \frac{48GD(\delta + G)}{\eta} \sum_{t=1}^{N} \beta_{1t} + \frac{108\eta^2 G^2(\delta + G)}{\delta} \sum_{t=1}^{N} \beta_{1t}^2, \tag{8}$$

which comes to our conclusion.

## B    PROOF FOR THE CONVERGENCE OF AMSGRAD-EG-DRD

In the $t$-th iteration of AMSGrad-EG-DRD, our update is as follows:

$$m_t = \beta_{1t}\hat{m}_{t-1} + (1 - \beta_{1t})g_{t-1}, \ v_t = \max(\beta_2\hat{v}_{t-1} + (1 - \beta_2)g_{t-1}^2, \hat{v}_{t-1})$$

$$H_t = \delta I + \mathrm{Diag}(\sqrt{v_t}), \ \hat{z}_t := (\hat{x}_t, \hat{y}_t) = (x_{t-1} + \eta \cdot (H_t^x)^{-1}m_t^x, y_{t-1} + \frac{\eta}{\sqrt{t}} \cdot (H_t^y)^{-1}m_t^y)$$

$$\hat{m}_t = \beta_{1t}m_t + (1 - \beta_{1t})\hat{g}_t, \ \hat{v}_t = \max(\beta_2 v_t + (1 - \beta_2)\hat{g}_t^2, v_t) \tag{9}$$

$$\hat{H}_t = \delta I + \mathrm{Diag}(\sqrt{\hat{v}_t}), \ z_t := (x_t, y_t) = (x_{t-1} + \eta \cdot (\hat{H}_t^x)^{-1}\hat{m}_t^x, y_{t-1} + \frac{\eta}{\sqrt{t}} \cdot (\hat{H}_t^y)^{-1}\hat{m}_t^y).$$

Most parts of this convergence proof are similar to the convergence proof of AMSGrad-EG. Lemma A.1 still holds.

**Lemma B.1.**

$$\|x_t - x^*\|_{\hat{H}_t^x}^2 \leqslant \|x_{t-1} - x^*\|_{\hat{H}_t^x}^2 - \|x_{t-1} - \hat{x}_t\|_{\hat{H}_t^x}^2 + \|\hat{x}_t - x_t\|_{\hat{H}_t^x}^2 + 2\langle \eta \cdot \hat{\varepsilon}_t^x, \hat{x}_t - x^* \rangle + 8\eta\beta_{1t}GD.$$

*Here,* $\hat{\varepsilon}_t^x = \hat{g}_t^x - V_x(\hat{z}_t) = \frac{1}{M_t}\sum_{i=1}^{M_t} V_x(\hat{z}_t; \hat{\xi}_i) - V_x(\hat{z}_t)$.

We can use the same technique of Lemma A.2 to prove it. Next, we obtain the next lemma.

**Lemma B.2.**

$$\|\hat{x}_t - x_t\|_{\hat{H}_t^x}^2 \leqslant 2\eta^2 \cdot \|((\hat{H}_t^x)^{-1} - (H_t^x)^{-1})m_t^x\|_{\hat{H}_t^x}^2 + \frac{16\eta^2 G^2 \beta_{1t}^2}{\delta} + \frac{12\eta^2 L^2}{\delta^2} \cdot \|\hat{z}_t - z_{t-1}\|_{\hat{H}_t}^2$$

$$+ 12\eta^2 \cdot \left( \|\hat{\varepsilon}_t^x\|_{(\hat{H}_t^x)^{-1}}^2 + \|\varepsilon_{t-1}^x\|_{(H_t^x)^{-1}}^2 \right).$$

*Here,* $\varepsilon_{t-1} = g_{t-1} - V(z_{t-1})$ *and* $\hat{\varepsilon}_t = \hat{g}_t - V(\hat{z}_t)$.

We can prove it by using the same technique as Lemma A.3. Since we have $\frac{12\eta^2 L^2}{\delta^2} \leqslant \frac{1}{2}$. Now, we can combine Lemma B.1 and Lemma B.2:

$$\|x_t - x^*\|_{\hat{H}_t^x}^2 \leqslant \|x_{t-1} - x^*\|_{\hat{H}_t^x}^2 - \|x_{t-1} - \hat{x}_t\|_{\hat{H}_t^x}^2 + 2\langle \eta \cdot \hat{\varepsilon}_t^x, \hat{x}_t - x^* \rangle + 8\eta\beta_{1t}GD$$

$$+ 2\eta^2 \cdot \|((\hat{H}_t^x)^{-1} - (H_t^x)^{-1})m_t^x\|_{\hat{H}_t^x}^2 + \frac{16\eta^2 G^2 \beta_{1t}^2}{\delta} + \frac{1}{2}\|\hat{z}_t - z_{t-1}\|_{\hat{H}_t}^2$$

$$+ 12\eta^2 \cdot \left( \|\hat{\varepsilon}_t^x\|_{(\hat{H}_t^x)^{-1}}^2 + \|\varepsilon_{t-1}^x\|_{(H_t^x)^{-1}}^2 \right),$$

After taking expectation and summation over $t = 1, 2, \ldots, N$, we obtain that:

$$\frac{1}{2}\sum_{t=1}^{N}\mathbb{E}\|x_{t-1} - \hat{x}_t\|_{\hat{H}_t^x}^2 \leqslant \sum_{t=1}^{N}\mathbb{E}\left[\|x_{t-1} - x^*\|_{\hat{H}_t^x}^2 - \|x_t - x^*\|_{\hat{H}_t^x}^2\right] + \sum_{t=1}^{N}\left[8\eta\beta_{1t}GD + \frac{16\eta^2 G^2\beta_{1t}^2}{\delta}\right]$$

$$+ 2\eta^2 \cdot \sum_{t=1}^{N}\mathbb{E}\|((\hat{H}_t^x)^{-1} - (H_t^x)^{-1})m_t^x\|_{\hat{H}_t^x}^2 + 12\eta^2 \cdot \sum_{t=1}^{N}\mathbb{E}\left[\|\hat{\varepsilon}_t^x\|_{(\hat{H}_t^x)^{-1}}^2 + \|\varepsilon_{t-1}^x\|_{(H_t^x)^{-1}}^2\right]$$

$$+ \frac{1}{2}\sum_{t=1}^{N}\mathbb{E}\|y_{t-1} - \hat{y}_t\|_{\hat{H}_t^y}^2.$$

$$(10)$$

Since $\|x_{t-1} - \hat{x}_t\|_{\hat{H}_t^x}^2 = \|\eta \cdot (H_t^x)^{-1}m_t^x\|_{\hat{H}_t^x}^2$ and $V_x(z_{t-1}) = g_{t-1}^x - \varepsilon_{t-1}^x = m_t^x - \beta_{1t}(\hat{m}_{t-1}^x - g_{t-1}^x) - \varepsilon_{t-1}^x$, we know that:

$$\|V_x(z_{t-1})\|_{(H_t^x)^{-1}}^2 = \|m_t^x - \beta_{1t}(\hat{m}_{t-1}^x - g_{t-1}^x) - \varepsilon_{t-1}^x\|_{(H_t^x)^{-1}}^2$$

$$\leqslant 3\left(\|m_t^x\|_{(H_t^x)^{-1}}^2 + \|\beta_{1t}(\hat{m}_{t-1}^x - g_{t-1}^x)\|_{(H_t^x)^{-1}}^2 + \|\varepsilon_{t-1}^x\|_{(H_t^x)^{-1}}^2\right)$$

$$= 3\left(\|\beta_{1t}(\hat{m}_{t-1}^x - g_{t-1}^x)\|_{(H_t^x)^{-1}}^2 + \|\varepsilon_{t-1}^x\|_{(H_t^x)^{-1}}^2\right) + 3 \cdot \|(H_t^x)^{-1}m_t^x\|_{\hat{H}_t^x}^2$$

$$\leqslant \frac{12G^2\beta_{1t}^2}{\delta} + 3\|\varepsilon_{t-1}^x\|_{(H_t^x)^{-1}}^2 + \frac{3}{\eta^2}\|x_{t-1} - \hat{x}_t\|_{\hat{H}_t^x}^2$$

$$(11)$$

After combining Equation (10) and Equation (11), it holds that:

$$\sum_{t=1}^{N}\mathbb{E}\|V_x(z_{t-1})\|_{(H_t^x)^{-1}}^2 \leqslant \frac{6}{\eta^2}\sum_{t=1}^{N}\mathbb{E}\left[\|x_{t-1} - x^*\|_{\hat{H}_t^x}^2 - \|x_t - x^*\|_{\hat{H}_t^x}^2\right] + \sum_{t=1}^{N}\left[\frac{48\beta_{1t}GD}{\eta} + \frac{108\eta^2 G^2\beta_{1t}^2}{\delta}\right]$$

$$+ 12\sum_{t=1}^{N}\mathbb{E}\|((\hat{H}_t^x)^{-1} - (H_t^x)^{-1})m_t^x\|_{\hat{H}_t^x}^2 + 75\sum_{t=1}^{N}\mathbb{E}\left[\|\hat{\varepsilon}_t^x\|_{(\hat{H}_t^x)^{-1}}^2 + \|\varepsilon_{t-1}^x\|_{(H_t^x)^{-1}}^2\right]$$

$$+ \frac{3}{\eta^2}\sum_{t=1}^{N}\mathbb{E}\|y_{t-1} - \hat{y}_t\|_{\hat{H}_t^y}^2.$$

$$(12)$$

In the following steps, we will upper bound the five terms above on the right side. Actually, the first four terms can be upper bounded by using the same technique in the convergence proof of AMSGrad-EG algorithm above.

**Lemma B.3.**
$$\sum_{t=1}^{N}\left[\|x_{t-1} - x^*\|_{\hat{H}_t^x}^2 - \|x_t - x^*\|_{\hat{H}_t^x}^2\right] \leqslant dD^2 \cdot (\delta + G),$$

*which is a constant.*

**Lemma B.4.**
$$\sum_{t=1}^{N}\|((H_t^x)^{-1} - (\hat{H}_t^x)^{-1})m_t\|_{\hat{H}_t^x}^2 \leqslant \frac{dG^2(\delta + G)}{\delta^2}.$$

Finally, the last term can be perfectly bounded by the $\mathcal{O}(1/\sqrt{t})$ decayed learning rate.

$$\sum_{t=1}^{N}\mathbb{E}\|y_{t-1} - \hat{y}_t\|_{\hat{H}_t^y}^2 \leqslant \sum_{t=1}^{N}\mathbb{E}\frac{\eta^2}{t}\|(\hat{H}_t^y)^{-1}m_t^y\|_{\hat{H}_t^y}^2$$

$$\leqslant \sum_{t=1}^{N}\mathbb{E}\frac{\eta^2}{t} \cdot \frac{G^2}{\delta} = \frac{\eta^2 G^2}{\delta}(1 + \log N). \quad (13)$$

To sum up, we eventually get the equation that:

$$\mathbb{E}\|V_x(z)\|_2^2 \leqslant \frac{(15 + 3\log N)dG^2(\delta + G)^2}{N\delta^2} + \frac{6dD^2(\delta + G)^2}{N\eta^2} + \frac{150\sigma^2(\delta + G)}{N\delta}\sum_{t=1}^{N}\frac{1}{M_t}$$

$$+ \frac{48GD(\delta + G)}{N\eta} \sum_{t=1}^{N} \beta_{1t} + \frac{108\eta^2 G^2(\delta + G)}{N\delta} \sum_{t=1}^{N} \beta_{1t}^2,$$

which comes to our conclusion.

## C MORE EXPERIMENTAL RESULTS

In this section, we further use experiments to verify the effectiveness of AMSGrad-EG and AMSGrad-EG-DRD algorithms proposed by us. Also, we show that the one-sided MVI condition is more feasible than standard MVI condition even in a more complicated setting. Here, we use DCGAN (Radford et al., 2015) on CIFAR10 dataset. We set our batch size as 100, learning rate as $1e$-4 and we compare AMSGrad-EG, AMSGrad-EG-DRD and SGDA by drawing their generated figures after 50, 100, 200 iterations in the following Figure 2. The two algorithms proposed by us again perform better than the non-adaptive SGDA. After training DCGAN on the CIFAR10 dataset with AMSGrad-EG optimizer, we plot the total MVI values $\langle -V(z_k), z_k - z^* \rangle$, $x$-sided MVI values $\langle -V_x(z_k), x_k - x^* \rangle$, and $y$-sided MVI values $\langle -V_y(z_k), y_k - y^* \rangle$ along the training trajectory as the following Figure 4.

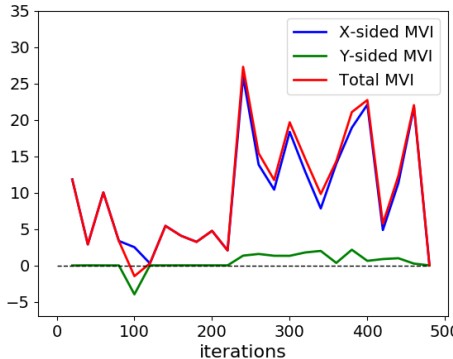

Figure 4: This figure shows the MVI values along the DCGAN's training trajectory. As we can see, the blue curve stays above $x$ axis in the experiment while the red curve does not.

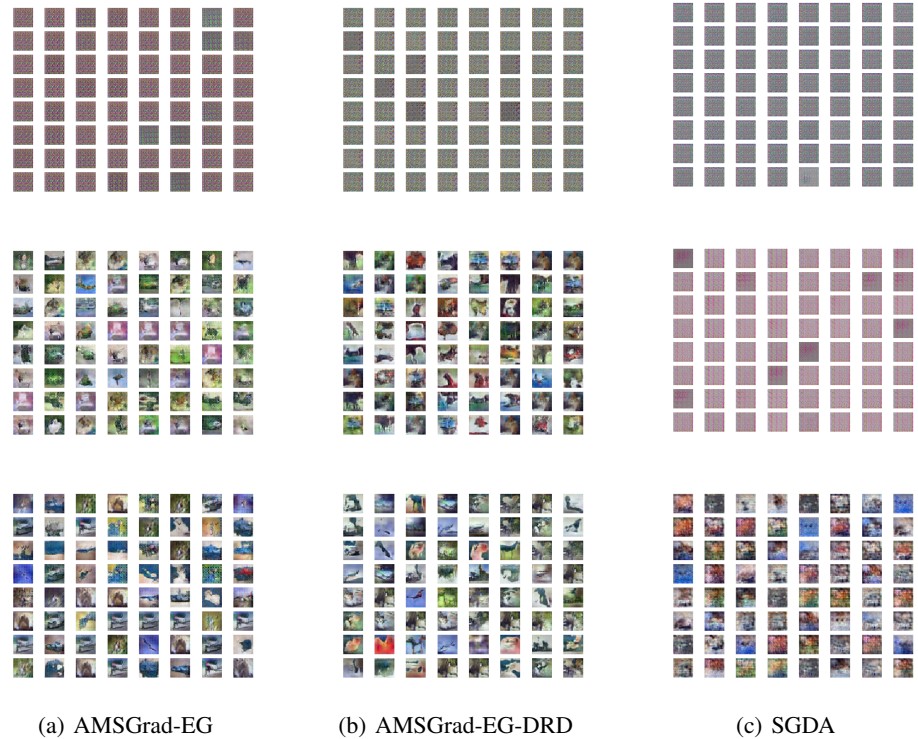

(a) AMSGrad-EG       (b) AMSGrad-EG-DRD       (c) SGDA

Figure 5: These are the generated CIFAR10 figures by the three algorithms after 50, 100, 200 iterations.

# D   STOCHASTIC EXTRA-GRADIENT AND ADAPTIVE EXTRA-GRADIENT ALGORITHMS

In a previous paper (Liu et al., 2020), the authors propose the Optimistic Gradient (OG) method and Optimistic AdaGrad (OAdagrad) method. In this section, we extend them to the **Extra-Gradient** type algorithms: Stochastic Extra-Gradient (SEG) and Adaptive Extra-Gradient (AEG). We put the result in the appendix because it is just a by-product of this research and not our main result.

Before we introduce our newly-proposed Adaptive Extra-Gradient method, we slightly modify the Stochastic Extra-Gradient (SEG) algorithm, by using different batch sizes in each iteration, as Algorithm 2 below.

---

**Algorithm 2** Stochastic Extra-Gradient (SEG) with batch size

---

**Input:** The initial state $\hat{z}_0 = z_0 = 0$, a constant learning rate $\eta$, a Stochastic First-order Oracle (SFO) $V(z; \xi)$, a sequence of batch sizes $\{m_k\}_{k \geqslant 1}$.
**Output:** $z_t$, $t$ is uniformly chosen from $\{0, 1, 2, \ldots, N-1\}$.
 1: **for** $k = 1, \ldots, N$ **do**
 2:     (Gradient Evaluation) $g_{k-1} = \frac{1}{m_k} \sum_{i=1}^{m_k} V(z_{k-1}; \xi_{k-1}^i)$.
 3:     (Shadow Update) $\hat{z}_k = \Pi_{\mathcal{Z}}[z_{k-1} + \eta \cdot g_{k-1}]$.
 4:     (Gradient Evaluation) $\hat{g}_{k-1} = \frac{1}{m_k} \sum_{i=1}^{m_k} V(\hat{z}_k; \hat{\xi}_{k-1}^i)$
 5:     (Real Update) $z_k = \Pi_{\mathcal{Z}}[z_{k-1} + \eta \cdot \hat{g}_{k-1}]$
 6: **end for**

---

Algorithm 3 is a basic idea on the design of AEG algorithm. Here, we use constant batch size $m$ and let $g_{k-1} = \frac{1}{m} \sum_{i=1}^{m} V(z_{k-1}, \xi_{k-1}^i), \hat{g}_{k-1} = \frac{1}{m} \sum_{i=1}^{m} V(\hat{z}_k, \hat{\xi}_{k-1}^i)$ be the estimated gradients on $z_{k-1}$ and $\hat{z}_k$ in the $k$-th iteration. Also, $\hat{g}_{1:k}$ is the concatenation of $\hat{g}_1, \ldots, \hat{g}_k$, and $\hat{g}_{1:k,i}$ is its $i$-th

row vector. Similarly, $g_{0:k}$ is the concatenation of $g_0, \ldots, g_k$, and $g_{0:k,i}$ is its $i$-th row vector. Note that all the matrices $H_k, S_k$ are diagonal so they don't require extra computation complexity.

---

**Algorithm 3** Adaptive Extra-Gradient (AEG)

---

**Input:** The initial state $\hat{z}_0 = z_0 = 0, H_0 = \hat{H}_0 = \delta I$, a constant learning rate $\eta$, a Stochastic First-order Oracle (SFO) $V(z; \xi)$, a constant batch size $m$.
**Output:** $z_t$, $t$ is uniformly chosen from $\{0, 1, 2, \ldots, N-1\}$.
 1: **for** $k = 1, \ldots, N$ **do**
 2:    (Gradient Evaluation) $g_{k-1} = \frac{1}{m} \sum_{i=1}^{m} V(z_{k-1}; \xi_{k-1}^i)$.
 3:    (Gradient Concatenation and Norm Calculation) Update $g_{0:k} = [g_{0:k-2}\ g_{k-1}], s_{k-1,i} = \| (g_{0:k-1,i}\ \hat{g}_{1:k-1,i}) \|_2$   $i = 1, 2, \ldots, d$ and $H_{k-1} = \delta I + \mathrm{diag}(s_{k-1})$.
 4:    (Shadow Update) $\hat{z}_k = z_{k-1} + \eta \cdot H_{k-1}^{-1} g_{k-1}$
 5:    (Gradient Evaluation) $\hat{g}_{k-1} = \frac{1}{m} \sum_{i=1}^{m} V(\hat{z}_k; \hat{\xi}_{k-1}^i)$.
 6:    (Gradient Concatenation and Norm Calculation) Update $\hat{g}_{1:k} = [\hat{g}_{1:k-1}\ \hat{g}_k], \hat{s}_{k-1,i} = \| (g_{0:k-1,i}\ \hat{g}_{1:k,i}) \|_2$   $i = 1, 2, \ldots, d$ and $S_{k-1} = \delta I + \mathrm{diag}(\hat{s}_{k-1})$.
 7:    (Real Update) $z_k = z_{k-1} + \eta \cdot S_{k-1}^{-1} \hat{g}_{k-1}$
 8: **end for**

---

Now, we introduce the following two theorems (Theorem D.1 and Theorem D.2), which compare the convergence rates of SEG and its adaptive variant AEG.

**Theorem D.1** (Convergence of SEG). *There is an algorithm (Stochastic Extra-Gradient), which given:*

- *A Stochastic First-order Oracle (SFO) access to the objective function $\phi(x, y) : \mathcal{X} \times \mathcal{Y} \to \mathbb{R}$ where $\mathcal{X} \subseteq \mathbb{R}^{n_1}, \mathcal{Y} \subseteq \mathbb{R}^{n_2}$, which is denoted as: $V(z; \xi)$ and it satisfies the following conditions:*

$$\mathbb{E}[V(z; \xi)] = V(z) := (-\nabla_x \phi, \nabla_y \phi) \ \text{and} \ \mathbb{E}\|V(z; \xi) - V(z)\|^2 \leqslant \sigma^2.$$

  *Here, $z := (x, y)$ and $\mathcal{Z} := \mathcal{X} \times \mathcal{Y}$. Also: the corresponding MVI function of $-V$ has a solution $z^* \in \mathcal{Z}$, which means $\langle -V(z), z - z^* \rangle \geqslant 0$ holds for $\forall z \in \mathcal{Z}$.*

- *A positive real $L$ such that $V$ is $L$-Lipschitz continuous with respect to $\| \cdot \|_2$.*

- *The learning rate $\eta$.*

- *An initial point $z_0 \in \mathcal{Z}$.*

- *The iteration number $N$.*

- *The sequence of batch sizes in each iteration $\{m_k\}_{k \geqslant 1}$.*

*Then, we output a result $z \in \mathcal{Z}$ which satisfies:*

$$\mathbb{E}[r_\eta^2(z)] \leqslant \frac{4\|z_0 - z^*\|^2}{N} + \frac{50\eta^2}{N} \sum_{k=0}^{N-1} \frac{\sigma^2}{m_k}$$

*where $r_\alpha(z) := \|z - \Pi_\mathcal{Z}(z + \alpha \cdot V(z))\|_2$ and $\eta \leqslant \frac{1}{4L}$. If $\mathcal{Z} = \mathbb{R}^{n_1 + n_2}$, then the projection operator $\Pi$ is the identity, and $r_\alpha(z) := \alpha \cdot \|V(z)\|_2$. The inequality above becomes:*

$$\mathbb{E}\|V(z)\|_2^2 \leqslant \frac{4\|z_0 - z^*\|^2}{\eta^2 N} + \frac{50}{N} \sum_{k=0}^{N-1} \frac{\sigma^2}{m_k}.$$

*Compared with Optimistic Stochastic Gradient (OSG) method proposed in (Liu et al., 2020), we can have a larger learning rate in this theorem.*

**Remark.** *Let $\eta = \frac{1}{4L}$. If the batch sizes are constant, which means $m_k = m$ for $\forall k$. In order to guarantee that $\mathbb{E}\|V(z)\|^2 \leqslant \varepsilon^2$, we have to make $m = O(1/\varepsilon^2)$ and $N = O(L^2/\varepsilon^2)$, and the total complexity is $\sum_{k=1}^{N} m_k = mN = O(L^2/\varepsilon^4)$. In another scenario where $m_k = k + 1$ is an increasing sequence, in order to guarantee that $\mathbb{E}\|V(z)\|^2 \leqslant \varepsilon^2$, we have to make $N = O(L^2/\varepsilon^2)$ and then the total complexity is $\sum_{k=1}^{N} m_k = \sum_{k=1}^{N} (k + 1) = O(L^4/\varepsilon^4)$.*

**Theorem D.2** (Convergence of AEG). *When our objective function $\phi(x, y)$ satisfies Assumption 1 and 2, as well as the bounded cumulative gradient condition, there exists an algorithm (AEG), given:*

- *A Stochastic First-order Oracle (SFO) access to the objective function $\phi(x, y) : \mathbb{R}^{n_1 + n_2} \to \mathbb{R}$, which is denoted as: $V(z; \xi)$ and it satisfies the following conditions:*

$$\mathbb{E}[V(z; \xi)] = V(z) := (-\nabla_x \phi, \nabla_y \phi) \ \ and \ \ \mathbb{E}\|V(z; \xi) - V(z)\|^2 \leqslant \sigma^2.$$

  *Here, $z := (x, y)$ and $\mathcal{Z} := \mathbb{R}^{n_1 + n_2} = \mathbb{R}^d$ where $d := n_1 + n_2$. Also: the corresponding MVI function of $V$ has a solution $z^* \in \mathcal{Z}$.*

- *Positive real numbers $G$, such that $\|V(z; \xi)\|_2 \leqslant G$ almost surely holds.*

- *A positive real $L$ such that $V$ is $L$-Lipschitz continuous with respect to $\|\cdot\|_2$.*

- *A universal constant $D > 0$ such that $\|z_k - z^*\|_2 \leqslant D$ and for all the points on the trajectory of our algorithm.*

- *An initial point $z_0 \in \mathcal{Z}$.*

- *The iteration number $N$.*

- *A constant batch size $m$.*

- *A constant $0 \leqslant \alpha \leqslant 1/2$ such that: the cumulative gradients are bounded as: $\|(g_{1:k,i} \ \hat{g}_{1:k,i})\|_2 \leqslant 2\delta \cdot k^\alpha$ for all $i, k$.*

- *The constant learning rate $\eta \leqslant \frac{\delta}{4L}$.*

*The output results $z_0, z_1, \ldots, z_{N-1}$ satisfies the following inequality:*

$$\frac{1}{N}\mathbb{E}\sum_{k=0}^{N-1}\|V(z_k)\|_{H_k^{-1}}^2 \leqslant \frac{16dD^2\delta/\eta^2 + 200d\delta + 40dG^2/\delta}{N^{1-\alpha}} + \frac{50\delta\sigma^2/m + 2G^2d/\delta}{N}.$$

If we switch the assumption of MVI condition (Assumption 2) to the one-sided MVI condition (Assumption 3), we can slightly modify the Adaptive Extra-Gradient (AEG) method to Adaptive Extra-Gradient with Dual Rate Decay (AEG-DRD), where the learning rate of $\{y_k\}$ variables set as $\frac{\eta}{k}$. In the Algorithm 4 below, $H_k = \text{Diag}(A_k, B_k)$ and $S_k = \text{Diag}(C_k, D_k)$ where for each $k \in \mathbb{N}$, $A_k, C_k \in \mathbb{R}^{n_1 \times n_1}$ and $B_k, D_k \in \mathbb{R}^{n_2 \times n_2}$. Also, $g_k = (g_k^x, g_k^y)$ and $\hat{g}_k = (\hat{g}_k^x, \hat{g}_k^y)$ where $g_k^x, \hat{g}_k^x \in \mathbb{R}^{n_1}$ and $g_k^y, \hat{g}_k^y \in \mathbb{R}^{n_2}$.

---

**Algorithm 4** Adaptive Extra-Gradient with Dual Rate Decay (AEG-DRD)

---

**Input:** The initial state $\hat{z}_0 = z_0 = 0, H_0 = \hat{H}_0 = \delta I$, a constant learning rate $\eta$, a Stochastic First-order Oracle (SFO) $V(z; \xi)$, a constant batch size $m$.
**Output:** $z_t$ where $t$ is uniformly chosen from $\{0, 1, 2, \ldots, N-1\}$.

1: **for** $k = 1, \ldots, N$ **do**
2:    (Gradient Evaluation) $g_{k-1} = \frac{1}{m}\sum_{i=1}^m V(z_{k-1}; \xi_{k-1}^i)$.
3:    (Gradient Concatenation and Norm Calculation) Update $g_{0:k} = [g_{0:k-2} \ g_{k-1}], s_{k-1,i} = \|(g_{0:k-1,i} \ \hat{g}_{1:k-1,i})\|_2 \ i = 1, 2, \ldots, d$ and $H_{k-1} = \delta I + \text{diag}(s_{k-1})$.
4:    (Shadow Update of $x$) $\hat{x}_k = x_{k-1} + \eta \cdot A_{k-1}^{-1} g_{k-1}^x$.
5:    (Shadow Update of $y$) $\hat{y}_k = y_{k-1} + \frac{\eta}{k} \cdot B_{k-1}^{-1} g_{k-1}^y$.
6:    (Gradient Evaluation) $\hat{g}_{k-1} = \frac{1}{m}\sum_{i=1}^m V(\hat{z}_k; \hat{\xi}_{k-1}^i)$.
7:    (Gradient Concatenation and Norm Calculation) Update $\hat{g}_{1:k} = [\hat{g}_{1:k-1} \ \hat{g}_k], \hat{s}_{k-1,i} = \|(g_{0:k-1,i} \ \hat{g}_{1:k,i})\|_2 \ i = 1, 2, \ldots, d$ and $S_{k-1} = \delta I + \text{diag}(\hat{s}_{k-1})$.
8:    (Real Update of $x$) $x_k = x_{k-1} + \eta \cdot C_{k-1}^{-1} \hat{g}_{k-1}^x$.
9:    (Real Update of $y$) $y_k = y_{k-1} + \frac{\eta}{k} \cdot D_{k-1}^{-1} \hat{g}_{k-1}^y$.
10: **end for**

---

**Theorem D.3** (Convergence of AEG-DRD). *When our objective function $\phi(x, y)$ satisfies Assumption 2 and 3, as well as the bounded cumulative gradient condition, there exists an algorithm (AEG-DRD), given:*

- *A Stochastic First-order Oracle (SFO) access to the objective function $\phi(x, y) : \mathbb{R}^{n_1+n_2} \to \mathbb{R}$, which is denoted as: $V(z; \xi)$ and it satisfies the following conditions:*

$$\mathbb{E}[V(z; \xi)] = V(z) := (-\nabla_x \phi, \nabla_y \phi) \ \text{and} \ \mathbb{E}\|V(z; \xi) - V(z)\|^2 \leqslant \sigma^2.$$

  *Here, $z := (x, y)$ and $\mathcal{Z} := \mathbb{R}^{n_1+n_2} = \mathbb{R}^d$ where $d := n_1 + n_2$. Also: the one-sided MVI function of $V$ has a solution $z^* \in \mathcal{Z}$.*

- *Positive real numbers $G$, such that $\|V(z; \xi)\|_2 \leqslant G$ almost surely holds.*

- *A positive real $L$ such that $V$ is $L$-Lipschitz continuous with respect to $\|\cdot\|_2$.*

- *A universal constant $D > 0$ such that $\|z_k - z^*\|_2 \leqslant D$ and for all the points on the trajectory of our algorithm.*

- *An initial point $z_0 \in \mathcal{Z}$.*

- *The iteration number $N$.*

- *A constant batch size $m$.*

- *A constant $0 \leqslant \alpha \leqslant 1/2$ such that: the cumulative gradients are bounded as: $\|(g_{1:k,i} \ \hat{g}_{1:k,i})\|_2 \leqslant 2\delta \cdot k^\alpha$ for all $i, k$.*

- *The constant learning rate $\eta \leqslant \frac{\delta}{4L}$.*

*The output results $z_0, z_1, \ldots, z_{N-1}$ satisfy the following inequality:*

$$\frac{1}{N} \mathbb{E} \sum_{k=1}^{N} \|V_x(z_{k-1})\|^2_{C_{k-1}^{-1}} \leqslant \frac{16dD^2\delta/\eta^2 + 200d\delta + 40dG^2/\delta}{N^{1-\alpha}} + \frac{24dG^2/\delta + 50\delta\sigma^2/m}{N}.$$

## D.1 PROOF OF THEOREM D.1

We recall that in the $t$-th iteration of Stochastic Extra-Gradient (SEG) algorithm with batch size, we have the following updates:

$$\hat{z}_t = \Pi_{\mathcal{Z}} \left[ z_{t-1} + \eta \cdot \frac{1}{m_t} \sum_{i=1}^{m_t} V(z_{t-1}; \xi_{t-1}^i) \right] := \Pi_{\mathcal{Z}} \left[ z_{t-1} + \eta \cdot g_{t-1} \right]$$

$$z_t = \Pi_{\mathcal{Z}} \left[ z_{t-1} + \eta \cdot \frac{1}{m_t} \sum_{i=1}^{m_t} V(\hat{z}_t; \hat{\xi}_{t-1}^i) \right] := \Pi_{\mathcal{Z}} \left[ z_{t-1} + \eta \cdot \hat{g}_{t-1} \right]$$

Before we start our proof, we introduce two simple properties of the projection operation $\Pi_{\mathcal{Z}}$ as follows, where $\mathcal{Z} \subseteq \mathbb{R}^d$ is closed and convex.

**Lemma D.1.**
*(1) $\|x - \Pi_{\mathcal{Z}}(x)\|^2 + \|\Pi_{\mathcal{Z}}(x) - z\|^2 \geqslant \|x - z\|^2$ and $\langle \Pi_{\mathcal{Z}}(x) - x, \Pi_{\mathcal{Z}}(x) - z \rangle \leqslant 0$ holds for $\forall x \in \mathbb{R}^d, z \in \mathcal{Z}$. Actually, the first inequality is a simple extension of the second one.*
*(2) The projection operator $\Pi_{\mathcal{Z}}$ is a compression, which means for $\forall x, y \in \mathbb{R}^d$, it holds that:*

$$\|\Pi_{\mathcal{Z}}(x) - \Pi_{\mathcal{Z}}(y)\|_2 \leqslant \|x - y\|_2.$$

Now, we can start our formal proof.

**Lemma D.2.**
$$\|z_k - z^*\|^2 \leqslant \|z_{k-1} - z^*\|^2 - (1 - 6\eta^2 L^2) \cdot \|z_{k-1} - \hat{z}_k\|^2 - \|z_k - \hat{z}_k\|^2$$
$$+ 6\eta^2 (\|\varepsilon_{k-1}\|^2 + \|\hat{\varepsilon}_{k-1}\|^2) + 2\langle \hat{z}_k - z^*, \eta \cdot \hat{\varepsilon}_{k-1} \rangle.$$

*Here, $\varepsilon_{k-1} = \frac{1}{m_k} \sum_{i=1}^{m_k} V(z_{k-1}; \xi_{k-1}^i) - V(z_{k-1})$ and $\hat{\varepsilon}_{k-1} = \frac{1}{m_k} \sum_{i=1}^{m_k} V(\hat{z}_k; \hat{\xi}_{k-1}^i) - V(\hat{z}_k)$.*

*Proof of Lemma D.2.* According to the Property (1) of Lemma D.1, we know that:

$$\|z_k - z^*\|^2 = \|\Pi_{\mathcal{Z}}[z_{k-1} + \eta \cdot \hat{g}_{k-1}] - z^*\|^2 \leqslant \|z_{k-1} + \eta \cdot \hat{g}_{k-1} - z^*\|^2 - \|z_{k-1} + \eta \cdot \hat{g}_{k-1} - z_k\|^2$$

$$= \|z_{k-1} - z^*\|^2 - \|z_{k-1} - z_k\|^2 + 2\langle \eta \cdot \hat{g}_{k-1}, z_k - z^* \rangle$$

$$= \|z_{k-1} - z^*\|^2 - \|z_{k-1} - \hat{z}_k + \hat{z}_k - z_k\|^2 + 2\langle \eta \cdot \hat{g}_{k-1}, \hat{z}_k - z^* \rangle + 2\langle \eta \cdot \hat{g}_{k-1}, z_k - \hat{z}_k \rangle$$

$$= \|z_{k-1} - z^*\|^2 - \|z_{k-1} - \hat{z}_k\|^2 - \|\hat{z}_k - z_k\|^2 - 2\langle z_{k-1} - \hat{z}_k, \hat{z}_k - z_k \rangle$$
$$+ 2\langle \eta \cdot (V(\hat{z}_k) + \hat{\varepsilon}_{k-1}), \hat{z}_k - z^* \rangle + 2\langle \eta \cdot \hat{g}_{k-1}, z_k - \hat{z}_k \rangle$$

$$= \|z_{k-1} - z^*\|^2 - \|z_{k-1} - \hat{z}_k\|^2 - \|\hat{z}_k - z_k\|^2 + 2\langle \hat{z}_k - z_k, \hat{z}_k - z_{k-1} - \eta \cdot \hat{g}_{k-1} \rangle$$
$$+ 2\langle \eta \cdot V(\hat{z}_k), \hat{z}_k - z^* \rangle + 2\langle \eta \cdot \hat{\varepsilon}_{k-1}, \hat{z}_k - z^* \rangle$$

Since $z^*$ is a solution of MVI inequality, we know that for $\forall z \in \mathcal{Z}$, it holds that $\langle V(z), z - z^* \rangle \leqslant 0$,. Therefore:

$$\langle V(\hat{z}_k), \hat{z}_k - z^* \rangle \leqslant 0.$$

Also, since $\hat{z}_k = \Pi_{\mathcal{Z}}[z_{k-1} + \eta \cdot g_{k-1}]$, according to the property (1) of Lemma D.1, we have:

$$\langle \hat{z}_k - z_k, \hat{z}_k - (z_{k-1} + \eta \cdot g_{k-1}) \rangle \leqslant 0.$$

After combining all the three inequalities above, we have:

$$\|z_k - z^*\|^2 \leqslant \|z_{k-1} - z^*\|^2 - \|z_{k-1} - \hat{z}_k\|^2 - \|\hat{z}_k - z_k\|^2 + 2\langle \eta \cdot \hat{\varepsilon}_{k-1}, \hat{z}_k - z^* \rangle$$
$$+ 2\langle \hat{z}_k - z_k, \hat{z}_k - z_{k-1} - \eta \cdot g_{k-1} + \eta \cdot g_{k-1} - \eta \cdot \hat{g}_{k-1} \rangle$$

$$\leqslant \|z_{k-1} - z^*\|^2 - \|z_{k-1} - \hat{z}_k\|^2 - \|\hat{z}_k - z_k\|^2 + 2\langle \eta \cdot \hat{\varepsilon}_{k-1}, \hat{z}_k - z^* \rangle$$
$$+ 2\langle \hat{z}_k - z_k, \eta \cdot g_{k-1} - \eta \cdot \hat{g}_{k-1} \rangle$$

$$\overset{(a)}{\leqslant} \|z_{k-1} - z^*\|^2 - \|z_{k-1} - \hat{z}_k\|^2 - \|\hat{z}_k - z_k\|^2 + 2\eta\|\hat{z}_k - z_k\| \cdot \|g_{k-1} - \hat{g}_{k-1}\|$$
$$+ 2\langle \eta \cdot \hat{\varepsilon}_{k-1}, \hat{z}_k - z^* \rangle$$

$$\overset{(b)}{\leqslant} \|z_{k-1} - z^*\|^2 - \|z_{k-1} - \hat{z}_k\|^2 - \|\hat{z}_k - z_k\|^2 + 2\eta^2\|g_{k-1} - \hat{g}_{k-1}\|^2$$
$$+ 2\langle \eta \cdot \hat{\varepsilon}_{k-1}, \hat{z}_k - z^* \rangle \tag{14}$$

Here, $(a)$ holds by Cauchy-Schwarz inequality, and $(b)$ holds because by using the property (2) of Lemma D.1, we know that

$$\|\hat{z}_k - z_k\| = \|\Pi_{\mathcal{Z}}[z_{k-1} + \eta \cdot g_{k-1}] - \Pi_{\mathcal{Z}}[z_{k-1} + \eta \cdot \hat{g}_{k-1}]\| \leqslant \|(z_{k-1} + \eta \cdot g_{k-1}) - (z_{k-1} + \eta \cdot \hat{g}_{k-1})\|$$
$$= \eta \cdot \|g_{k-1} - \hat{g}_{k-1}\|.$$

Now, we focus on dealing with $\|g_{k-1} - \hat{g}_{k-1}\|$. In fact:

$$\|g_{k-1} - \hat{g}_{k-1}\|^2 \overset{(c)}{=} \|V(z_{k-1}) - V(\hat{z}_k) + \varepsilon_{k-1} - \hat{\varepsilon}_{k-1}\|^2$$

$$\overset{(d)}{\leqslant} 3\left(\|V(z_{k-1}) - V(\hat{z}_k)\|^2 + \|\varepsilon_{k-1}\|^2 + \|\hat{\varepsilon}_{k-1}\|^2\right) \tag{15}$$

$$\overset{(e)}{\leqslant} 3L^2\|z_{k-1} - \hat{z}_k\|^2 + 3(\|\varepsilon_{k-1}\|^2 + \|\hat{\varepsilon}_{k-1}\|^2)$$

Here, $(c)$ holds by the definition of $\varepsilon_{k-1}, \hat{\varepsilon}_{k-1}$. $(d)$ holds because by Cauchy Inequality, $\|x + y + z\|^2 \leqslant 3(\|x\|^2 + \|y\|^2 + \|z\|^2)$ always holds. $(e)$ holds because we've assumed that $V(z)$ is $L$-Lipschitz continuous under $\|\cdot\|_2$ norm. Combine Equation (14) and (15), we obtain that:

$$\|z_k - z^*\|^2 \leqslant \|z_{k-1} - z^*\|^2 - (1 - 6\eta^2 L^2)\|z_{k-1} - \hat{z}_k\|^2 - \|\hat{z}_k - z_k\|^2 + 6\eta^2 \cdot (\|\varepsilon_{k-1}\|^2 + \|\hat{\varepsilon}_{k-1}\|^2)$$
$$+ 2\langle \eta \cdot \hat{\varepsilon}_{k-1}, \hat{z}_k - z^* \rangle,$$

which is exactly what we want to prove in this lemma. $\square$

Now, we come back to our Theorem D.1. Notice that

$$r_\eta(z_k)^2 = \|z_k - \Pi_{\mathcal{Z}}[z_k + \eta \cdot V(z_k)]\|^2 = \|z_k - \hat{z}_{k+1} + \hat{z}_{k+1} - \Pi_{\mathcal{Z}}[z_k + \eta \cdot V(z_k)]\|^2$$

$$\overset{(a)}{\leqslant} 2\left(\|z_k - \hat{z}_{k+1}\|^2 + \|\hat{z}_{k+1} - \Pi_{\mathcal{Z}}[z_k + \eta \cdot V(z_k)]\|^2\right)$$

$$= 2\left(\|z_k - \hat{z}_{k+1}\|^2 + \|\Pi_{\mathcal{Z}}[z_k + \eta \cdot g_k] - \Pi_{\mathcal{Z}}[z_k + \eta \cdot V(z_k)]\|^2\right) \tag{16}$$

$$\overset{(b)}{\leqslant} 2\|z_k - \hat{z}_{k+1}\|^2 + 2\eta^2 \cdot \|\varepsilon_k\|^2$$

Here, $(a)$ comes from Cauchy Inequality and $(b)$ holds because of the property (2) of Lemma D.1.

*Proof of Theorem D.1.* Put $k = 1, 2, \ldots, N$ in Lemma D.2 and add them up, and we obtain that:

$$\|z_N - z^*\|^2 \leqslant \|z_0 - z^*\|^2 - (1 - 6\eta^2 L^2) \cdot \sum_{k=0}^{N-1} \|z_k - \hat{z}_{k+1}\|^2 - \sum_{k=1}^{N} \|z_k - \hat{z}_k\|^2$$

$$+ 6\eta^2 \cdot \sum_{k=0}^{N-1} \left( \|\varepsilon_k\|^2 + \|\hat{\varepsilon}_k\|^2 \right) + 2 \sum_{k=1}^{N} \langle \hat{z}_k - z^*, \eta \cdot \hat{\varepsilon}_{k-1} \rangle.$$

Therefore, after combining with Equation (16), we have:

$$\sum_{k=0}^{N-1} r_\eta(z_k)^2 \leqslant 2 \sum_{k=0}^{N-1} \|z_k - \hat{z}_{k+1}\|^2 + 2\eta^2 \cdot \sum_{k=0}^{N-1} \|\varepsilon_k\|^2$$

$$\leqslant \frac{2}{1 - 6\eta^2 L^2} \left( \|z_0 - z^*\|^2 + 6\eta^2 \cdot \sum_{k=0}^{N-1} \left( \|\varepsilon_k\|^2 + \|\hat{\varepsilon}_k\|^2 \right) + 2 \sum_{k=1}^{N} \langle \hat{z}_k - z^*, \eta \cdot \hat{\varepsilon}_{k-1} \rangle \right)$$

$$+ 2\eta^2 \cdot \sum_{k=0}^{N-1} \|\varepsilon_k\|^2$$

Since $\eta \leqslant \frac{1}{4L}$, we have $1 - 6\eta^2 L^2 \geqslant \frac{1}{2}$. After taking expectation on the inequality above, it holds that:

$$\frac{1}{N} \sum_{k=0}^{N-1} \mathbb{E}[r_\eta(z_k)^2] \leqslant \frac{4\|z_0 - z^*\|^2}{N} + \frac{48\eta^2}{N} \sum_{k=0}^{N-1} \frac{\sigma^2}{m_k} + \frac{2\eta^2}{N} \sum_{k=0}^{N-1} \frac{\sigma^2}{m_k}$$

$$= \frac{4\|z_0 - z^*\|^2}{N} + \frac{50\eta^2}{N} \sum_{k=0}^{N-1} \frac{\sigma^2}{m_k}$$

which comes to our conclusion. Here, we use the fact that $\mathbb{E}\|\varepsilon_k\|^2 \leqslant \frac{\sigma^2}{m_k}, \mathbb{E}\|\hat{\varepsilon}_k\|^2 \leqslant \frac{\sigma^2}{m_k}$ and $\mathbb{E}\langle \hat{z}_k - z^*, \eta \cdot \hat{\varepsilon}_{k-1} \rangle = 0$. □

## D.2 PROOF OF THEOREM D.2

We recall that in the $t$-th iteration of Adaptive Extra-Gradient, our update is as follows:

$$\hat{z}_t = z_{t-1} + \eta \cdot H_{t-1}^{-1} g_{t-1}$$

$$z_t = z_{t-1} + \eta \cdot S_{t-1}^{-1} \hat{g}_t.$$

Now we begin our proof by introducing several lemmas.

**Lemma D.3.**

$$\|z_k - z^*\|_{S_{k-1}}^2 \leqslant \|z_{k-1} - z^*\|_{S_{k-1}}^2 - \|z_{k-1} - \hat{z}_k\|_{S_{k-1}}^2 + \|z_k - \hat{z}_k\|_{S_{k-1}}^2 + 2\langle \eta \cdot \hat{\varepsilon}_k, \hat{z}_k - z^* \rangle.$$

Here, $\hat{\varepsilon}_k = \hat{g}_k - V(\hat{z}_k) = \frac{1}{m} \sum_{i=1}^{m} V(\hat{z}_k; \hat{\xi}_i) - V(\hat{z}_k)$.

*Proof of Lemma D.3.* According to our update rule, we know that:

$$\|z_k - z^*\|_{S_{k-1}}^2 = \|z_{k-1} + \eta \cdot S_{k-1}^{-1} \hat{g}_k - z^*\|_{S_{k-1}}^2$$

$$= \|z_{k-1} + \eta \cdot S_{k-1}^{-1} \hat{g}_k - z^*\|_{S_{k-1}}^2 - \|z_{k-1} + \eta \cdot S_{k-1}^{-1} \hat{g}_k - z_k\|_{S_{k-1}}^2$$

$$= \|z_{k-1} - z^*\|_{S_{k-1}}^2 - \|z_{k-1} - z_k\|_{S_{k-1}}^2 + 2\langle \eta \cdot \hat{g}_k, z_k - z^* \rangle$$

$$= \|z_{k-1} - z^*\|_{S_{k-1}}^2 - \|z_{k-1} - \hat{z}_k + \hat{z}_k - z_k\|_{S_{k-1}}^2 + 2\langle \eta \cdot \hat{g}_k, z_k - \hat{z}_k \rangle + 2\langle \eta \cdot \hat{g}_k, \hat{z}_k - z^* \rangle$$

$$= \|z_{k-1} - z^*\|_{S_{k-1}}^2 - \|z_{k-1} - \hat{z}_k\|_{S_{k-1}}^2 - \|\hat{z}_k - z_k\|_{S_{k-1}}^2 - 2\langle S_{k-1}(z_{k-1} - \hat{z}_k), \hat{z}_k - z_k \rangle$$

$$\quad + 2\langle \eta \cdot \hat{g}_k, z_k - \hat{z}_k \rangle + 2\langle \eta \cdot \hat{g}_k, \hat{z}_k - z^* \rangle$$

$$= \|z_{k-1} - z^*\|_{S_{k-1}}^2 - \|z_{k-1} - \hat{z}_k\|_{S_{k-1}}^2 - \|\hat{z}_k - z_k\|_{S_{k-1}}^2 + 2\langle \eta \cdot \hat{g}_k, \hat{z}_k - z^* \rangle$$

$$\quad + 2\langle z_k - \hat{z}_k, S_{k-1}(z_{k-1} - \hat{z}_k + \eta \cdot S_{k-1}^{-1} \hat{g}_k) \rangle$$

$$= \|z_{k-1} - z^*\|_{S_{k-1}}^2 - \|z_{k-1} - \hat{z}_k\|_{S_{k-1}}^2 - \|\hat{z}_k - z_k\|_{S_{k-1}}^2 + 2\langle \eta \cdot \hat{g}_k, \hat{z}_k - z^* \rangle$$

$$\quad + 2\langle z_k - \hat{z}_k, S_{k-1}(z_k - \hat{z}_k) \rangle$$

$$= \|z_{k-1} - z^*\|_{S_{k-1}}^2 - \|z_{k-1} - \hat{z}_k\|_{S_{k-1}}^2 + \|\hat{z}_k - z_k\|_{S_{k-1}}^2 + 2\langle \eta \cdot \hat{g}_k, \hat{z}_k - z^* \rangle$$

Notice that $\hat{g}_k = V(\hat{z}_k) + \hat{\varepsilon}_k$. Since $z^*$ is a solution of MVI which means $\langle V(z), z - z^* \rangle \leqslant 0$ holds for $\forall z \in \mathcal{Z}$, so $\langle V(\hat{z}_k), \hat{z}_k - z^* \rangle \leqslant 0$. Therefore:

$$\langle \eta \cdot \hat{g}_k, \hat{z}_k - z^* \rangle = \langle \eta \cdot (V(\hat{z}_k) + \hat{\varepsilon}_k), \hat{z}_k - z^* \rangle \leqslant \langle \eta \cdot \hat{\varepsilon}_k, \hat{z}_k - z^* \rangle.$$

Combine it with the inequality above, we obtain that:

$$\|z_k - z^*\|^2_{S_{k-1}} \leqslant \|z_{k-1} - z^*\|^2_{S_{k-1}} - \|z_{k-1} - \hat{z}_k\|^2_{S_{k-1}} + \|\hat{z}_k - z_k\|^2_{S_{k-1}} + 2\langle \eta \cdot \hat{\varepsilon}_k, \hat{z}_k - z^* \rangle,$$

which comes to our conclusion. $\qquad\square$

In the lemma above, it's worth mentioned that the expectation of $\langle \eta \cdot \hat{\varepsilon}_k, \hat{z}_k - z^* \rangle$ is 0. In the next lemma, we are going to deal with the upper bound of $\|\hat{z}_k - z_k\|^2_{S_{k-1}}$. Before that, we notice the following fact:

$$H_0 \preceq S_0 \preceq H_1 \preceq S_1 \preceq \ldots \preceq H_N \preceq S_N.$$

**Lemma D.4.**

$$\|z_k - \hat{z}_k\|^2_{S_{k-1}} \leqslant \frac{6\eta^2 L^2}{\delta^2} \|z_{k-1} - \hat{z}_k\|^2_{S_{k-1}} + 6\eta^2 \left( \|\varepsilon_{k-1}\|^2_{H^{-1}_{k-1}} + \|\hat{\varepsilon}_{k-1}\|^2_{S^{-1}_{k-1}} \right)$$
$$+ 2\eta^2 \cdot \|(S^{-1}_{k-1} - H^{-1}_{k-1})g_{k-1}\|^2_{S_{k-1}}$$

*Proof of Lemma D.4.* Since $\hat{z}_k = z_{k-1} + \eta \cdot H^{-1}_{k-1}g_{k-1}$ and $z_k = z_{k-1} + \eta \cdot S^{-1}_{k-1}\hat{g}_k$, we have:

$$\|z_k - \hat{z}_k\|^2_{S_{k-1}} = \eta^2 \cdot \|S^{-1}_{k-1}\hat{g}_k - H^{-1}_{k-1}g_{k-1}\|^2_{S_{k-1}} = \eta^2 \cdot \|S^{-1}_{k-1}\hat{g}_k - S^{-1}_{k-1}g_{k-1} + S^{-1}_{k-1}g_{k-1} - H^{-1}_{k-1}g_{k-1}\|^2_{S_{k-1}}$$

$$\overset{(a)}{\leqslant} 2\eta^2 \cdot \left( \|S^{-1}_{k-1}(\hat{g}_k - g_{k-1})\|^2_{S_{k-1}} + \|(S^{-1}_{k-1} - H^{-1}_{k-1})g_{k-1}\|^2_{S_{k-1}} \right)$$

$$= 2\eta^2 \cdot \left( \|\hat{g}_k - g_{k-1}\|^2_{S^{-1}_{k-1}} + \|(S^{-1}_{k-1} - H^{-1}_{k-1})g_{k-1}\|^2_{S_{k-1}} \right)$$

$$= 2\eta^2 \cdot \|V(\hat{z}_k) - V(z_{k-1}) + \hat{\varepsilon}_{k-1} - \varepsilon_{k-1}\|^2_{S^{-1}_{k-1}} + 2\eta^2 \cdot \|(S^{-1}_{k-1} - H^{-1}_{k-1})g_{k-1}\|^2_{S_{k-1}}$$

$$\overset{(b)}{\leqslant} 6\eta^2 \cdot \left( \|V(\hat{z}_k) - V(z_{k-1})\|^2_{S^{-1}_{k-1}} + \|\hat{\varepsilon}_{k-1}\|^2_{S^{-1}_{k-1}} + \|\varepsilon_{k-1}\|^2_{S^{-1}_{k-1}} \right)$$
$$+ 2\eta^2 \cdot \|(S^{-1}_{k-1} - H^{-1}_{k-1})g_{k-1}\|^2_{S_{k-1}}$$

$$\overset{(c)}{\leqslant} 6\eta^2 \cdot \left( \frac{L^2}{\delta^2} \|\hat{z}_k - z_{k-1}\|^2_{S_{k-1}} + \|\hat{\varepsilon}_{k-1}\|^2_{S^{-1}_{k-1}} + \|\varepsilon_{k-1}\|^2_{H^{-1}_{k-1}} \right)$$
$$+ 2\eta^2 \cdot \|(S^{-1}_{k-1} - H^{-1}_{k-1})g_{k-1}\|^2_{S_{k-1}},$$

which comes to our conclusion. Here, $(a)$ holds since for any norm $\|\cdot\|_A$, we have $\|x + y\|^2_A \leqslant (\|x\|_A + \|y\|_A)^2 \leqslant 2(\|x\|^2_A + \|y\|^2_A)$, and $(b)$ holds for the similar reason. $(c)$ holds because of the following two reasons:
(1) Since $V$ is $L$-Lipschitz continuous, and $\delta I \preceq S_{k-1}$, we have:

$$\|V(\hat{z}_k) - V(z_{k-1})\|^2_{S^{-1}_{k-1}} \leqslant \frac{1}{\delta} \|V(\hat{z}_k) - V(z_{k-1})\|^2_2 \leqslant \frac{L^2}{\delta} \|\hat{z}_k - z_{k-1}\|^2 \leqslant \frac{L^2}{\delta^2} \|\hat{z}_k - z_{k-1}\|^2_{S_{k-1}}.$$

(2) Since $H_{k-1} \preceq S_{k-1}$, we have $S^{-1}_{k-1} \preceq H^{-1}_{k-1}$. Therefore,

$$\|\varepsilon_{k-1}\|^2_{S^{-1}_{k-1}} \leqslant \|\varepsilon_{k-1}\|^2_{H^{-1}_{k-1}}.$$

$\qquad\square$

Since $\eta \leqslant \frac{\delta}{4L}$, then $\frac{6\eta^2 L^2}{\delta^2} \leqslant \frac{1}{2}$. According to Lemma D.3 and Lemma D.4, we have:

$$\|z_k - z^*\|^2_{S_{k-1}} \leqslant \|z_{k-1} - z^*\|^2_{S_{k-1}} - \|z_{k-1} - \hat{z}_k\|^2_{S_{k-1}} + 2\langle \eta \cdot \hat{\varepsilon}_{k-1}, \hat{z}_k - z^* \rangle + \frac{1}{2} \|z_{k-1} - \hat{z}_k\|^2_{S_{k-1}}$$

$$+ 6\eta^2 \left( \|\varepsilon_{k-1}\|^2_{H^{-1}_{k-1}} + \|\hat{\varepsilon}_{k-1}\|^2_{S^{-1}_{k-1}} \right) + 2\eta^2 \cdot \|(S^{-1}_{k-1} - H^{-1}_{k-1})g_{k-1}\|^2_{S_{k-1}}$$

which means:

$$\frac{1}{2}\|z_{k-1} - \hat{z}_k\|^2_{S_{k-1}} \leqslant \|z_{k-1} - z^*\|^2_{S_{k-1}} - \|z_k - z^*\|^2_{S_{k-1}} + 2\langle \eta \cdot \hat{\varepsilon}_{k-1}, \hat{z}_k - z^* \rangle$$
$$+ 6\eta^2 \left( \|\varepsilon_{k-1}\|^2_{H^{-1}_{k-1}} + \|\hat{\varepsilon}_{k-1}\|^2_{S^{-1}_{k-1}} \right) + 2\eta^2 \cdot \|(S^{-1}_{k-1} - H^{-1}_{k-1})g_{k-1}\|^2_{S_{k-1}} \tag{17}$$

Notice that, $\|z_{k-1} - \hat{z}_k\|^2_{S_{k-1}} = \|\eta \cdot H^{-1}_{k-1}g_{k-1}\|^2_{S_{k-1}}$. Therefore,

$$\|g_{k-1}\|^2_{S^{-1}_{k-1}} = \|S^{-1}_{k-1}g_{k-1}\|^2_{S_{k-1}} = \|H^{-1}_{k-1}g_{k-1} + (S^{-1}_{k-1} - H^{-1}_{k-1})g_{k-1}\|^2_{S_{k-1}}$$
$$\leqslant 2 \left( \|H^{-1}_{k-1}g_{k-1}\|^2_{S_{k-1}} + \|(S^{-1}_{k-1} - H^{-1}_{k-1})g_{k-1}\|^2_{S_{k-1}} \right) \tag{18}$$
$$= \frac{2}{\eta^2} \cdot \|z_{k-1} - \hat{z}_k\|^2_{S_{k-1}} + 2 \cdot \|(S^{-1}_{k-1} - H^{-1}_{k-1})g_{k-1}\|^2_{S_{k-1}}$$

Combine Equation (17) and (18), we obtain that:

$$\|g_{k-1}\|^2_{S^{-1}_{k-1}} \leqslant \frac{4}{\eta^2} \cdot (\|z_{k-1} - z^*\|^2_{S_{k-1}} - \|z_k - z^*\|^2_{S_{k-1}}) + \frac{8}{\eta}\langle \hat{\varepsilon}_{k-1}, \hat{z}_k - z^* \rangle$$
$$+ 24 \cdot \left( \|\varepsilon_{k-1}\|^2_{H^{-1}_{k-1}} + \|\hat{\varepsilon}_{k-1}\|^2_{S^{-1}_{k-1}} \right) + 10 \cdot \|(S^{-1}_{k-1} - H^{-1}_{k-1})g_{k-1}\|^2_{S_{k-1}} \tag{19}$$

In the inequality above, the expectation of $\langle \hat{\varepsilon}_{k-1}, \hat{z}_k - z^* \rangle$ is 0, which can be ignored under expectation. Since we are going to do the summation over $k = 1, 2, \ldots, N$ in the future, in the following lemmas, we analyze the upper bounds of $\sum_{k=1}^{N}(\|z_{k-1} - z^*\|^2_{S_{k-1}} - \|z_k - z^*\|^2_{S_{k-1}})$, $\sum_{k=1}^{N} \|(S^{-1}_{k-1} - H^{-1}_{k-1})g_{k-1}\|^2_{S_{k-1}}$, and $\sum_{k=1}^{N}(\|\varepsilon_{k-1}\|^2_{H^{-1}_{k-1}} + \|\hat{\varepsilon}_{k-1}\|^2_{S^{-1}_{k-1}})$

**Lemma D.5.**

$$\sum_{k=1}^{N} \left( \|z_{k-1} - z^*\|^2_{S_{k-1}} - \|z_k - z^*\|^2_{S_{k-1}} \right) \leqslant 2dD^2\delta \cdot N^{\alpha}.$$

*Proof of Lemma D.5.* Notice that

$$\sum_{k=1}^{N} \left( \|z_{k-1} - z^*\|^2_{S_{k-1}} - \|z_k - z^*\|^2_{S_{k-1}} \right) \leqslant \|z_0 - z^*\|^2_{S_0} + \sum_{k=1}^{N-1} \left( \|z_k - z^*\|^2_{S_k} - \|z_k - z^*\|^2_{S_{k-1}} \right)$$

$$= \|z_0 - z^*\|^2_{S_0} + \sum_{k=1}^{N-1} \left[ (z_k - z^*)^\top (S_k - S_{k-1}) \cdot (z_k - z^*) \right]$$

$$\leqslant D^2 \cdot \mathrm{tr}(S_0) + \sum_{k=1}^{N-1} D^2 \cdot (\mathrm{tr}(S_k) - \mathrm{tr}(S_{k-1})) = D^2 \cdot \mathrm{tr}(S_{N-1}) < D^2 \cdot 2d\delta N^{\alpha},$$

which comes to our conclusion. □

**Lemma D.6.**

$$\sum_{k=1}^{N} \|(S^{-1}_{k-1} - H^{-1}_{k-1})g_{k-1}\|^2_{S_{k-1}} \leqslant \frac{2dG^2 \cdot N^{\alpha}}{\delta}.$$

*Proof of Lemma D.6.* For any $\delta \leqslant x < y$, we notice that:

$$y \left( \frac{1}{x} - \frac{1}{y} \right)^2 = \frac{(y-x)^2}{x^2 y} < \frac{y-x}{x^2} \leqslant \frac{y-x}{\delta^2}. \tag{20}$$

Therefore, we have:

$$\sum_{k=1}^{N} \|(S_{k-1}^{-1} - H_{k-1}^{-1})g_{k-1}\|_{S_{k-1}}^2 \leqslant \sum_{k=1}^{N} g_{k-1}^{\top}(H_{k-1}^{-1} - S_{k-1}^{-1})S_{k-1}(H_{k-1}^{-1} - S_{k-1}^{-1})g_{k-1}$$

$$\leqslant \sum_{k=1}^{N} G^2 \cdot \text{tr}\left((H_{k-1}^{-1} - S_{k-1}^{-1})S_{k-1}(H_{k-1}^{-1} - S_{k-1}^{-1})\right)$$

$$\overset{(a)}{\leqslant} \sum_{k=1}^{N} \frac{G^2}{\delta^2} \cdot [\text{tr}(S_{k-1}) - \text{tr}(H_{k-1})]$$

$$\leqslant \frac{G^2}{\delta^2} \cdot \text{tr}(S_{N-1}) < \frac{G^2}{\delta^2} \cdot 2d\delta \cdot N^\alpha = \frac{2dG^2 \cdot N^\alpha}{\delta}.$$

Here, $(a)$ holds because of Equation (20). $\qquad\square$

**Lemma D.7.**
$$\mathbb{E}\sum_{k=1}^{N}\left(\|\varepsilon_{k-1}\|_{H_{k-1}^{-1}}^2 + \|\hat{\varepsilon}_{k-1}\|_{S_{k-1}^{-1}}^2\right) \leqslant \frac{\delta\sigma^2}{m} + 4d\delta \cdot N^\alpha.$$

The proof of this lemma can be found in (Duchi et al., 2011). Combine Lemma D.5, D.6, D.7 with Equation (19), we obtain that:

$$\mathbb{E}\sum_{k=1}^{N}\|g_{k-1}\|_{S_{k-1}^{-1}}^2 \leqslant \frac{4}{\eta^2} \cdot 2dD^2\delta N^\alpha + 24 \cdot \left(\frac{\delta\sigma^2}{m} + 4d\delta \cdot N^\alpha\right) + 10 \cdot \frac{2dG^2 \cdot N^\alpha}{\delta}.$$

Now, we replace the $g_{k-1}$ with the true gradient $V(z_{k-1})$ as follows:

$$\mathbb{E}\sum_{k=1}^{N}\|V(z_{k-1})\|_{H_{k-1}^{-1}}^2 = \mathbb{E}\sum_{k=1}^{N}\|g_{k-1} + \varepsilon_{k-1}\|_{H_{k-1}^{-1}}^2 \leqslant 2 \cdot \mathbb{E}\sum_{k=1}^{N}\|g_{k-1}\|_{H_{k-1}^{-1}}^2 + 2 \cdot \mathbb{E}\sum_{k=1}^{N}\|\varepsilon_{k-1}\|_{H_{k-1}^{-1}}^2$$

$$\overset{(a)}{\leqslant} 2 \cdot \mathbb{E}\sum_{k=1}^{N}\left[\|g_{k-1}\|_{S_{k-1}^{-1}}^2 + g_{k-1}^{\top}(H_{k-1}^{-1} - S_{k-1}^{-1}) \cdot g_{k-1}\right] + 2 \cdot \left(\frac{\delta\sigma^2}{m} + 4d\delta \cdot N^\alpha\right)$$

$$\leqslant 2 \cdot \mathbb{E}\sum_{k=1}^{N}\|g_{k-1}\|_{S_{k-1}^{-1}}^2 + 2 \cdot \left(\frac{\delta\sigma^2}{m} + 4d\delta \cdot N^\alpha\right) + 2G^2 \cdot \sum_{k=1}^{N}\left[\text{tr}(H_{k-1}^{-1}) - \text{tr}(S_{k-1}^{-1})\right]$$

$$\leqslant \frac{8}{\eta^2} \cdot 2dD^2\delta N^\alpha + 50 \cdot \left(\frac{\delta\sigma^2}{m} + 4d\delta \cdot N^\alpha\right) + 20 \cdot \frac{2dG^2 \cdot N^\alpha}{\delta} + 2G^2 \cdot \text{tr}(H_0^{-1})$$

$$= \frac{8}{\eta^2} \cdot 2dD^2\delta N^\alpha + 50 \cdot \left(\frac{\delta\sigma^2}{m} + 4d\delta \cdot N^\alpha\right) + 20 \cdot \frac{2dG^2 \cdot N^\alpha}{\delta} + \frac{2G^2 d}{\delta}.$$

Here, $(a)$ uses the conclusion of Lemma D.7. Finally, we multiplies $\frac{1}{N}$ on both sides and finish the proof:

$$\frac{1}{N}\mathbb{E}\sum_{k=0}^{N-1}\|V(z_k)\|_{H_k^{-1}}^2 \leqslant \frac{16dD^2\delta/\eta^2 + 200d\delta + 40dG^2/\delta}{N^{1-\alpha}} + \frac{50\delta\sigma^2/m + 2G^2 d/\delta}{N}.$$

### D.3 PROOF OF THEOREM D.3

In this proof, we are still using the Adaptive Extra-Gradient (AEG) algorithm, which its $t$-th iteration is:

$$\hat{z}_t = z_{t-1} + \eta \cdot H_{t-1}^{-1}g_{t-1}$$

$$z_t = z_{t-1} + \eta \cdot S_{t-1}^{-1}\hat{g}_t.$$

For simplicity, we split the update of $x$ variable and $y$ variable. Recall that $H_k = \text{Diag}(A_k, B_k)$ and $S_k = \text{Diag}(C_k, D_k)$ where for each $k \in \mathbb{N}$, $A_k, C_k \in \mathbb{R}^{n_1 \times n_1}$ and $B_k, D_k \in \mathbb{R}^{n_2 \times n_2}$. Also, we

denote $g_k = (g_k^x, g_k^y)$ and $\hat{g}_k = (\hat{g}_k^x, \hat{g}_k^y)$ where $g_k^x, \hat{g}_k^x \in \mathbb{R}^{n_1}$ and $g_k^y, \hat{g}_k^y \in \mathbb{R}^{n_2}$. Then, the update above can be rewritten as:

$$\hat{x}_t = x_{t-1} + \eta \cdot A_{t-1}^{-1} g_{t-1}^x$$
$$\hat{y}_t = y_{t-1} + \frac{\eta}{t} \cdot B_{t-1}^{-1} g_{t-1}^y$$
$$x_t = x_{t-1} + \eta \cdot C_{t-1}^{-1} \hat{g}_t^x$$
$$y_t = y_{t-1} + \frac{\eta}{t} \cdot D_{t-1}^{-1} \hat{g}_t^y.$$

Similar to Lemma D.3, we only focus on the $x$-s, and then we can get our upper bound for $\|x_k - x^*\|_{C_{k-1}}^2$, which is the first step of our proof.

**Lemma D.8.**

$$\|x_k - x^*\|_{C_{k-1}}^2 \leqslant \|x_{k-1} - x^*\|_{C_{k-1}}^2 - \|x_{k-1} - \hat{x}_k\|_{C_{k-1}}^2 + \|x_k - \hat{x}_k\|_{C_{k-1}}^2 + 2\langle \eta \cdot \hat{\varepsilon}_{k-1}^x, \hat{x}_k - x^* \rangle.$$

*Proof of Lemma D.8.* According to the update rule of $x_k, \hat{x}_k$, we know that:

$$\begin{aligned}
\|x_k - x^*\|_{C_{k-1}}^2 &= \|x_{k-1} + \eta \cdot C_{k-1}^{-1} \hat{g}_k^x - x^*\|_{C_{k-1}}^2 \\
&= \|x_{k-1} + \eta \cdot C_{k-1}^{-1} \hat{g}_k^x - x^*\|_{C_{k-1}}^2 - \|x_{k-1} + \eta \cdot C_{k-1}^{-1} \hat{g}_k^x - x_k\|_{C_{k-1}}^2 \\
&= \|x_{k-1} - x^*\|_{C_{k-1}}^2 - \|x_{k-1} - x_k\|_{C_{k-1}}^2 + 2\langle \eta \cdot \hat{g}_k^x, x_k - x^* \rangle \\
&= \|x_{k-1} - x^*\|_{C_{k-1}}^2 - \|x_{k-1} - \hat{x}_k + \hat{x}_k - x_k\|_{C_{k-1}}^2 + 2\langle \eta \cdot \hat{g}_k^x, x_k - \hat{x}_k \rangle + 2\langle \eta \cdot \hat{g}_k^x, \hat{x}_k - x^* \rangle \\
&= \|x_{k-1} - x^*\|_{C_{k-1}}^2 - \|x_{k-1} - \hat{x}_k\|_{C_{k-1}}^2 - \|\hat{x}_k - x_k\|_{C_{k-1}}^2 - 2\langle C_{k-1}(x_{k-1} - \hat{x}_k), \hat{x}_k - x_k \rangle \\
&\quad + 2\langle \eta \cdot \hat{g}_k^x, x_k - \hat{x}_k \rangle + 2\langle \eta \cdot \hat{g}_k^x, \hat{x}_k - x^* \rangle \\
&= \|x_{k-1} - x^*\|_{C_{k-1}}^2 - \|x_{k-1} - \hat{x}_k\|_{C_{k-1}}^2 - \|\hat{x}_k - x_k\|_{C_{k-1}}^2 + 2\langle \eta \cdot \hat{g}_k^x, \hat{x}_k - x^* \rangle \\
&\quad + 2\langle x_k - \hat{x}_k, C_{k-1}(x_{k-1} - \hat{x}_k + \eta \cdot C_{k-1}^{-1} \hat{g}_k^x) \rangle \\
&= \|x_{k-1} - x^*\|_{C_{k-1}}^2 - \|x_{k-1} - \hat{x}_k\|_{C_{k-1}}^2 - \|\hat{x}_k - x_k\|_{C_{k-1}}^2 + 2\langle \eta \cdot \hat{g}_k^x, \hat{x}_k - x^* \rangle \\
&\quad + 2\langle x_k - \hat{x}_k, C_{k-1}(x_k - \hat{x}_k) \rangle \\
&= \|x_{k-1} - x^*\|_{C_{k-1}}^2 - \|x_{k-1} - \hat{x}_k\|_{C_{k-1}}^2 + \|\hat{x}_k - x_k\|_{C_{k-1}}^2 + 2\langle \eta \cdot \hat{g}_k^x, \hat{x}_k - x^* \rangle
\end{aligned}$$

Notice that $\hat{g}_k = V(\hat{z}_k) + \hat{\varepsilon}_{k-1} \Rightarrow \hat{g}_k^x = V_x(\hat{z}_k) + \hat{\varepsilon}_{k-1}^x$. Since $z^*$ satisfies the one-side MVI condition, therefore $\langle V_x(z), x - x^* \rangle \leqslant 0$ holds for $\forall z = (x, y) \in \mathcal{Z}$, so $\langle V_x(\hat{z}_k), \hat{x}_k - x^* \rangle \leqslant 0$. Therefore:

$$\langle \eta \cdot \hat{g}_k^x, \hat{x}_k - x^* \rangle = \langle \eta \cdot (V_x(\hat{z}_k) + \hat{\varepsilon}_{k-1}^x), \hat{x}_k - x^* \rangle \leqslant \langle \eta \cdot \hat{\varepsilon}_{k-1}^x, \hat{x}_k - x^* \rangle.$$

Combine it with the inequality above, we obtain that:

$$\|x_k - x^*\|_{C_{k-1}}^2 \leqslant \|x_{k-1} - x^*\|_{C_{k-1}}^2 - \|x_{k-1} - \hat{x}_k\|_{C_{k-1}}^2 + \|\hat{x}_k - x_k\|_{C_{k-1}}^2 + 2\langle \eta \cdot \hat{\varepsilon}_{k-1}^x, \hat{x}_k - x^* \rangle,$$

which comes to our conclusion. $\square$

Notice that the term $2\langle \eta \cdot \hat{\varepsilon}_{k-1}^x, \hat{x}_k - x^* \rangle$ has zero mean, which can be ignored when taking expectation. In the next step, we upper bound the $\|\hat{x}_k - x_k\|_{C_{k-1}}^2$ term. Also, we notice the following facts:

$$A_0 \preceq C_0 \preceq A_1 \preceq C_1 \preceq \ldots \preceq A_N \preceq C_N,$$
$$B_0 \preceq D_0 \preceq B_1 \preceq D_1 \preceq \ldots \preceq B_N \preceq D_N.$$

**Lemma D.9.**

$$\begin{aligned}
\|x_k - \hat{x}_k\|_{C_{k-1}}^2 &\leqslant \frac{6\eta^2 L^2}{\delta^2} \left( \|x_{k-1} - \hat{x}_k\|_{C_{k-1}}^2 + \|y_{k-1} - \hat{y}_k\|_{D_{k-1}}^2 \right) + 6\eta^2 \left( \|\varepsilon_{k-1}\|_{H_{k-1}^{-1}}^2 + \|\hat{\varepsilon}_{k-1}\|_{S_{k-1}^{-1}}^2 \right) \\
&\quad + 2\eta^2 \cdot \|(C_{k-1}^{-1} - A_{k-1}^{-1}) g_{k-1}^x\|_{C_{k-1}}^2
\end{aligned}$$

*Proof of Lemma D.9.* Since $\hat{x}_k = x_{k-1} + \eta \cdot A_{k-1}^{-1} g_{k-1}^x$ and $x_k = x_{k-1} + \eta \cdot C_{k-1}^{-1} \hat{g}_k^x$, we have:

$$\|x_k - \hat{x}_k\|_{C_{k-1}}^2 = \eta^2 \cdot \|C_{k-1}^{-1} \hat{g}_k^x - A_{k-1}^{-1} g_{k-1}^x\|_{C_{k-1}}^2 = \eta^2 \cdot \|C_{k-1}^{-1} \hat{g}_k^x - C_{k-1}^{-1} g_{k-1}^x + C_{k-1}^{-1} g_{k-1}^x - A_{k-1}^{-1} g_{k-1}^x\|_{C_{k-1}}^2$$

$$
\begin{aligned}
&\overset{(a)}{\leqslant} 2\eta^2 \cdot \left( \|C_{k-1}^{-1}(\hat{g}_k^x - g_{k-1}^x)\|_{C_{k-1}}^2 + \|(C_{k-1}^{-1} - A_{k-1}^{-1})g_{k-1}^x\|_{C_{k-1}}^2 \right) \\
&= 2\eta^2 \cdot \left( \|\hat{g}_k^x - g_{k-1}^x\|_{C_{k-1}^{-1}}^2 + \|(C_{k-1}^{-1} - A_{k-1}^{-1})g_{k-1}^x\|_{C_{k-1}}^2 \right) \\
&\overset{(b)}{\leqslant} 2\eta^2 \cdot \left( \|\hat{g}_k - g_{k-1}\|_{S_{k-1}^{-1}}^2 + \|(C_{k-1}^{-1} - A_{k-1}^{-1})g_{k-1}^x\|_{C_{k-1}}^2 \right) \\
&= 2\eta^2 \cdot \|V(\hat{z}_k) - V(z_{k-1}) + \hat{\varepsilon}_{k-1} - \varepsilon_{k-1}\|_{S_{k-1}^{-1}}^2 + 2\eta^2 \cdot \|(C_{k-1}^{-1} - A_{k-1}^{-1})g_{k-1}^x\|_{C_{k-1}}^2 \\
&\overset{(c)}{\leqslant} 6\eta^2 \cdot \left( \|V(\hat{z}_k) - V(z_{k-1})\|_{S_{k-1}^{-1}}^2 + \|\hat{\varepsilon}_{k-1}\|_{S_{k-1}^{-1}}^2 + \|\varepsilon_{k-1}\|_{S_{k-1}^{-1}}^2 \right) \\
&\quad + 2\eta^2 \cdot \|(C_{k-1}^{-1} - A_{k-1}^{-1})g_{k-1}^x\|_{C_{k-1}}^2 \\
&\overset{(d)}{\leqslant} 6\eta^2 \cdot \left( \frac{L^2}{\delta^2}\|\hat{z}_k - z_{k-1}\|_{S_{k-1}}^2 + \|\hat{\varepsilon}_{k-1}\|_{S_{k-1}^{-1}}^2 + \|\varepsilon_{k-1}\|_{H_{k-1}^{-1}}^2 \right) \\
&\quad + 2\eta^2 \cdot \|(C_{k-1}^{-1} - A_{k-1}^{-1})g_{k-1}^x\|_{C_{k-1}}^2 \\
&\overset{(e)}{=} 6\eta^2 \cdot \left( \frac{L^2}{\delta^2}\|\hat{x}_k - x_{k-1}\|_{C_{k-1}}^2 + \frac{L^2}{\delta^2}\|\hat{y}_k - y_{k-1}\|_{D_{k-1}}^2 + \|\hat{\varepsilon}_{k-1}\|_{S_{k-1}^{-1}}^2 + \|\varepsilon_{k-1}\|_{H_{k-1}^{-1}}^2 \right) \\
&\quad + 2\eta^2 \cdot \|(C_{k-1}^{-1} - A_{k-1}^{-1})g_{k-1}^x\|_{C_{k-1}}^2
\end{aligned}
\tag{21}
$$

which comes to our conclusion. Here, $(a)$ holds since for any norm $\|\cdot\|_A$, we have $\|x + y\|_A^2 \leqslant (\|x\|_A + \|y\|_A)^2 \leqslant 2(\|x\|_A^2 + \|y\|_A^2)$, and $(c)$ holds for the similar reason. $(b)$ holds because $\|\hat{g}_k - g_{k-1}\|_{S_{k-1}^{-1}}^2 = \|\hat{g}_k^x - g_{k-1}^x\|_{C_{k-1}^{-1}}^2 + \|\hat{g}_k^y - g_{k-1}^y\|_{D_{k-1}^{-1}}^2 \geqslant \|\hat{g}_k^x - g_{k-1}^x\|_{C_{k-1}^{-1}}^2$, and $(e)$ holds for the same reason. $(d)$ holds because of the following two reasons:
(1) Since $V$ is $L$-Lipschitz continuous, and $\delta I \preceq S_{k-1}$, we have:

$$
\|V(\hat{z}_k) - V(z_{k-1})\|_{S_{k-1}^{-1}}^2 \leqslant \frac{1}{\delta}\|V(\hat{z}_k) - V(z_{k-1})\|_2^2 \leqslant \frac{L^2}{\delta}\|\hat{z}_k - z_{k-1}\|^2 \leqslant \frac{L^2}{\delta^2}\|\hat{z}_k - z_{k-1}\|_{S_{k-1}}^2.
$$

(2) Since $H_{k-1} \preceq S_{k-1}$, we have $S_{k-1}^{-1} \preceq H_{k-1}^{-1}$. Therefore,

$$
\|\varepsilon_{k-1}\|_{S_{k-1}^{-1}}^2 \leqslant \|\varepsilon_{k-1}\|_{H_{k-1}^{-1}}^2.
$$

$\square$

Since we require our learning rate $\eta \leqslant \frac{\delta}{4L}$, the coefficient above $\frac{6\eta^2 L^2}{\delta^2} < \frac{1}{2}$. After combining Lemma D.8 and Lemma D.9, we obtain that:

$$
\|x_k - x^*\|_{C_{k-1}}^2 \leqslant \|x_{k-1} - x^*\|_{C_{k-1}}^2 - \frac{1}{2}\|x_{k-1} - \hat{x}_k\|_{C_{k-1}}^2 + \frac{1}{2}\|y_{k-1} - \hat{y}_k\|_{D_{k-1}}^2 + 2\langle \eta \cdot \hat{\varepsilon}_{k-1}^x, \hat{x}_k - x^* \rangle \\
+ 6\eta^2 \left( \|\varepsilon_{k-1}\|_{H_{k-1}^{-1}}^2 + \|\hat{\varepsilon}_{k-1}\|_{S_{k-1}^{-1}}^2 \right) + 2\eta^2 \cdot \|(C_{k-1}^{-1} - A_{k-1}^{-1})g_{k-1}^x\|_{C_{k-1}}^2,
\tag{22}
$$

which means:

$$
\frac{1}{2}\|x_{k-1} - \hat{x}_k\|_{C_{k-1}}^2 \leqslant \|x_{k-1} - x^*\|_{C_{k-1}}^2 - \|x_k - x^*\|_{C_{k-1}}^2 + \frac{1}{2}\|y_{k-1} - \hat{y}_k\|_{D_{k-1}}^2 + 2\langle \eta \cdot \hat{\varepsilon}_{k-1}^x, \hat{x}_k - x^* \rangle \\
+ 6\eta^2 \left( \|\varepsilon_{k-1}\|_{H_{k-1}^{-1}}^2 + \|\hat{\varepsilon}_{k-1}\|_{S_{k-1}^{-1}}^2 \right) + 2\eta^2 \cdot \|(C_{k-1}^{-1} - A_{k-1}^{-1})g_{k-1}^x\|_{C_{k-1}}^2,
\tag{23}
$$

Notice that, $\|x_{k-1} - \hat{x}_k\|_{C_{k-1}}^2 = \|\eta \cdot A_{k-1}^{-1}g_{k-1}^x\|_{C_{k-1}}^2$. Therefore,

$$
\begin{aligned}
\|g_{k-1}^x\|_{C_{k-1}^{-1}}^2 = \|C_{k-1}^{-1}g_{k-1}^x\|_{C_{k-1}}^2 &= \|A_{k-1}^{-1}g_{k-1}^x + (C_{k-1}^{-1} - A_{k-1}^{-1})g_{k-1}^x\|_{C_{k-1}}^2 \\
&\leqslant 2\left( \|A_{k-1}^{-1}g_{k-1}^x\|_{C_{k-1}}^2 + \|(C_{k-1}^{-1} - A_{k-1}^{-1})g_{k-1}^x\|_{C_{k-1}}^2 \right) \\
&= \frac{2}{\eta^2} \cdot \|x_{k-1} - \hat{x}_k\|_{C_{k-1}}^2 + 2 \cdot \|(C_{k-1}^{-1} - A_{k-1}^{-1})g_{k-1}^x\|_{C_{k-1}}^2.
\end{aligned}
\tag{24}
$$

Combining Equation (23) and Equation (24), we get:

$$\|g_{k-1}^x\|_{C_{k-1}^{-1}}^2 \leqslant \frac{4}{\eta^2}\left(\|x_{k-1}-x^*\|_{C_{k-1}}^2 - \|x_k-x^*\|_{C_{k-1}}^2\right) + \frac{2}{\eta^2}\|y_{k-1}-\hat{y}_k\|_{D_{k-1}}^2 + \frac{8}{\eta^2}\langle \eta \cdot \hat{\varepsilon}_{k-1}^x, \hat{x}_k - x^*\rangle$$
$$+ 24\left(\|\varepsilon_{k-1}\|_{H_{k-1}^{-1}}^2 + \|\hat{\varepsilon}_{k-1}\|_{S_{k-1}^{-1}}^2\right) + 10 \cdot \|(C_{k-1}^{-1} - A_{k-1}^{-1})g_{k-1}^x\|_{C_{k-1}}^2,$$

(25)

Then, we take summation over $k = 1, 2, \ldots, N$ and take expectation, and we can obtain that:

$$\mathbb{E}\sum_{k=1}^N \|g_{k-1}^x\|_{C_{k-1}^{-1}}^2 \leqslant \frac{4}{\eta^2}\mathbb{E}\sum_{k=1}^N\left(\|x_{k-1}-x^*\|_{C_{k-1}}^2 - \|x_k-x^*\|_{C_{k-1}}^2\right) + \frac{2}{\eta^2}\mathbb{E}\sum_{k=1}^N\|y_{k-1}-\hat{y}_k\|_{D_{k-1}}^2$$

$$+ 24 \cdot \mathbb{E}\sum_{k=1}^N\left(\|\varepsilon_{k-1}\|_{H_{k-1}^{-1}}^2 + \|\hat{\varepsilon}_{k-1}\|_{S_{k-1}^{-1}}^2\right) + 10 \cdot \mathbb{E}\sum_{k=1}^N\|(C_{k-1}^{-1} - A_{k-1}^{-1})g_{k-1}^x\|_{C_{k-1}}^2.$$

(26)

In the following steps, we will upper bound the four terms above on the right side. The third term $\mathbb{E}\sum_{k=1}^N\left(\|\varepsilon_{k-1}\|_{H_{k-1}^{-1}}^2 + \|\hat{\varepsilon}_{k-1}\|_{S_{k-1}^{-1}}^2\right)$ can be upper bounded by using Lemma D.7:

$$\mathbb{E}\sum_{k=1}^N\left(\|\varepsilon_{k-1}\|_{H_{k-1}^{-1}}^2 + \|\hat{\varepsilon}_{k-1}\|_{S_{k-1}^{-1}}^2\right) \leqslant \frac{\delta\sigma^2}{m} + 4d\delta \cdot N^\alpha.$$

(27)

The fourth term $\mathbb{E}\sum_{k=1}^N \|(C_{k-1}^{-1} - A_{k-1}^{-1})g_{k-1}^x\|_{C_{k-1}}^2$ can be upper bounded by using Lemma D.6:

$$\mathbb{E}\sum_{k=1}^N \|(C_{k-1}^{-1} - A_{k-1}^{-1})g_{k-1}^x\|_{C_{k-1}}^2 \leqslant \mathbb{E}\sum_{k=1}^N \|(S_{k-1}^{-1} - H_{k-1}^{-1})g_{k-1}\|_{S_{k-1}}^2 \leqslant \frac{2dG^2 \cdot N^\alpha}{\delta}.$$

For the first term $\mathbb{E}\sum_{k=1}^N\left(\|x_{k-1}-x^*\|_{C_{k-1}}^2 - \|x_k-x^*\|_{C_{k-1}}^2\right)$, we can use the similar techniques as Lemma D.5.

**Lemma D.10.**

$$\sum_{k=1}^N\left(\|x_{k-1}-x^*\|_{C_{k-1}}^2 - \|x_k-x^*\|_{C_{k-1}}^2\right) \leqslant 2dD^2\delta \cdot N^\alpha.$$

*Proof of Lemma D.10.* Notice that

$$\sum_{k=1}^N\left(\|x_{k-1}-x^*\|_{C_{k-1}}^2 - \|x_k-x^*\|_{C_{k-1}}^2\right) \leqslant \|x_0-x^*\|_{C_0}^2 + \sum_{k=1}^{N-1}\left(\|x_k-x^*\|_{C_k}^2 - \|x_k-x^*\|_{C_{k-1}}^2\right)$$

$$= \|x_0-x^*\|_{C_0}^2 + \sum_{k=1}^{N-1}\left[(x_k-x^*)^\top(C_k - C_{k-1})\cdot(x_k-x^*)\right]$$

$$\leqslant D^2 \cdot \text{tr}(C_0) + \sum_{k=1}^{N-1} D^2 \cdot (\text{tr}(C_k) - \text{tr}(C_{k-1})) = D^2 \cdot \text{tr}(C_{N-1}) < D^2 \cdot \text{tr}(S_{N-1}) \leqslant D^2 \cdot 2d\delta N^\alpha,$$

which comes to our conclusion. $\qquad\square$

For the second term $\mathbb{E}\sum_{k=1}^N\|y_{k-1}-\hat{y}_k\|_{D_{k-1}}^2$, notice that:

$$\sum_{k=1}^N\|y_{k-1}-\hat{y}_k\|_{D_{k-1}}^2 = \sum_{k=1}^N \frac{\eta^2}{k^2}\cdot\|B_{k-1}^{-1}g_{k-1}^y\|_{D_{k-1}}^2 = \sum_{k=1}^N \frac{\eta^2}{k^2}\cdot g_{k-1}^{y\top}B_{k-1}^{-1}D_{k-1}B_{k-1}^{-1}g_{k-1}^y$$

$$\leqslant \sum_{k=1}^N \frac{\eta^2}{k^2}\cdot G^2\text{tr}(B_{k-1}^{-1}D_{k-1}B_{k-1}^{-1}) < \sum_{k=1}^N \frac{\eta^2}{k^2}G^2 \cdot \frac{2d\delta k^\alpha}{\delta^2}$$

$$= \frac{2\eta^2 dG^2}{\delta}\sum_{k=1}^N \frac{1}{k^{2-\alpha}}$$

Here, we notice that:

$$\sum_{k=1}^{N} \frac{1}{k^{2-\alpha}} < 1 + \int_{1}^{N} \frac{1}{x^{2-\alpha}} dx < 1 + \int_{1}^{N} \frac{1}{x^{3/2}} dx = 1 + 2\left(1 - \frac{1}{\sqrt{N}}\right) < 3.$$

Therefore:

$$\sum_{k=1}^{N} \|y_{k-1} - \hat{y}_k\|_{D_{k-1}}^2 < \frac{6\eta^2 dG^2}{\delta}.$$

According to Equation (26):

$$
\begin{aligned}
\mathbb{E}\sum_{k=1}^{N} \|g_{k-1}^x\|_{C_{k-1}^{-1}}^2 &\leqslant \frac{4}{\eta^2} \cdot 2dD^2\delta \cdot N^\alpha + \frac{2}{\eta^2} \cdot \frac{6\eta^2 dG^2}{\delta} + 24 \cdot \left(\frac{\delta\sigma^2}{m} + 4d\delta \cdot N^\alpha\right) + 10 \cdot \frac{2dG^2 \cdot N^\alpha}{\delta} \\
&= N^\alpha \cdot \left(\frac{8dD^2\delta}{\eta^2} + 96d\delta + \frac{20dG^2}{\delta}\right) + \frac{12dG^2}{\delta} + \frac{24\delta\sigma^2}{m}.
\end{aligned}
$$

(28)

Finally, we replace the $g_{k-1}^x$ above with the actual gradient $V_x(z_{k-1})$. Since:

$$\|V_x(z_{k-1})\|_{C_{k-1}^{-1}}^2 \leqslant \left(\|g_{k-1}^x\|_{C_{k-1}^{-1}} + \|\varepsilon_{k-1}^x\|_{C_{k-1}^{-1}}\right)^2 \leqslant 2\left(\|g_{k-1}^x\|_{C_{k-1}^{-1}}^2 + \|\varepsilon_{k-1}^x\|_{C_{k-1}^{-1}}^2\right),$$

we can upper bound the target term $\mathbb{E}\sum_{k=1}^{N} \|V_x(z_{k-1})\|_{C_{k-1}^{-1}}^2$:

$$
\begin{aligned}
\mathbb{E}\sum_{k=1}^{N} \|V_x(z_{k-1})\|_{C_{k-1}^{-1}}^2 &\leqslant \mathbb{E}\sum_{k=1}^{N} 2\left(\|g_{k-1}^x\|_{C_{k-1}^{-1}}^2 + \|\varepsilon_{k-1}^x\|_{C_{k-1}^{-1}}^2\right) \leqslant 2\mathbb{E}\sum_{k=1}^{N} \|g_{k-1}^x\|_{C_{k-1}^{-1}}^2 + 2\mathbb{E}\sum_{k=1}^{N} \|\varepsilon_{k-1}^x\|_{C_{k-1}^{-1}}^2 \\
&\overset{(a)}{\leqslant} N^\alpha \cdot \left(\frac{16dD^2\delta}{\eta^2} + 192d\delta + \frac{40dG^2}{\delta}\right) + \frac{24dG^2}{\delta} + \frac{48\delta\sigma^2}{m} + 2\mathbb{E}\sum_{k=1}^{N} \left(\|\varepsilon_{k-1}\|_{H_{k-1}^{-1}}^2 + \|\hat{\varepsilon}_{k-1}\|_{S_{k-1}^{-1}}^2\right) \\
&\overset{(b)}{\leqslant} N^\alpha \cdot \left(\frac{16dD^2\delta}{\eta^2} + 192d\delta + \frac{40dG^2}{\delta}\right) + \frac{24dG^2}{\delta} + \frac{48\delta\sigma^2}{m} + 2\left(\frac{\delta\sigma^2}{m} + 4d\delta \cdot N^\alpha\right) \\
&= N^\alpha \cdot \left(\frac{16dD^2\delta}{\eta^2} + 200d\delta + \frac{40dG^2}{\delta}\right) + \frac{24dG^2}{\delta} + \frac{50\delta\sigma^2}{m},
\end{aligned}
$$

(29)

which means that:

$$\frac{1}{N}\mathbb{E}\sum_{k=1}^{N} \|V_x(z_{k-1})\|_{C_{k-1}^{-1}}^2 \leqslant \frac{16dD^2\delta/\eta^2 + 200d\delta + 40dG^2/\delta}{N^{1-\alpha}} + \frac{24dG^2/\delta + 50\delta\sigma^2/m}{N},$$

which comes to our conclusion.

# E   PSEUDO-ALGORITHM OF AMSGRAD-EG-DRD

---

**Algorithm 5** Extra Gradient AMSGrad with Dual Rate Decay

---

**Input:** The initial state $z_0 = m_0 = v_0 = 0$, a constant learning rate $\eta$, a Stochastic First-order Oracle (SFO) $V(z;\xi)$, momentum parameters $\beta_{1t}, \beta_2$, a sequence of batch sizes $\{M_k\}$.

**Output:** $z_t$ with $t$ uniformly chosen from $\{0, 1, \ldots, N-1\}$.

1: **for** $k = 1, \ldots, N$ **do**
2:     (Gradient Evaluation 1) $g_{k-1} = \frac{1}{m} \sum_{i=1}^{m} V(z_{k-1}; \xi_{k-1}^i)$.
3:     (Momentum Update 1)$m_k = \beta_{1k}\hat{m}_{k-1} + (1 - \beta_{1k})g_{k-1}$
4:     (Velocity Update 1) $v_k = \max(\beta_2\hat{v}_{k-1} + (1 - \beta_2)g_{k-1}^2, \hat{v}_{k-1})$, $H_k = \delta I + \text{Diag}(\sqrt{v_k})$.
5:     (Shadow Update) $\hat{x}_k = x_{k-1} + \eta \cdot (H_k^x)^{-1} m_k^x$,
         $\hat{y}_k = y_{k-1} + \frac{\eta}{\sqrt{k}} (H_k^y)^{-1} m_k^y$, $\hat{z}_k = (\hat{x}_k, \hat{y}_k)$.
6:     (Gradient Evaluation 2) $\hat{g}_k = \frac{1}{M_k} \sum_{i=1}^{M_k} V(\hat{z}_k; \xi_k^i)$.
7:     (Momentum Update 2) $\hat{m}_k = \beta_{1k}m_k + (1 - \beta_{1k})\hat{g}_k$.
8:     (Velocity Update 2) $\hat{v}_k = \max(\beta_2 v_k + (1 - \beta_2)\hat{g}_k^2, v_k)$, $\hat{H}_k = \delta I + \text{Diag}(\sqrt{\hat{v}_k})$
9:     (Real Update) $z_k = z_{k-1} + \eta \cdot \hat{H}_k^{-1}\hat{m}_k$.
10:     (Real Update) $x_k = x_{k-1} + \eta \cdot \left(\hat{H}_k^x\right)^{-1} \hat{m}_k^x$,
         $y_k = y_{k-1} + \frac{\eta}{\sqrt{k}} \left(\hat{H}_k^y\right)^{-1} \hat{m}_k^y$, $z_k = (x_k, y_k)$.
11: **end for**

---

