# OpenReview forum: "On the One-sided Convergence of Adam-type Algorithms in Non-convex Non-concave Min-max Optimization"
_ICLR.cc/2022/Conference — ICLR 2022 Submitted_

### Official Review · Reviewer_M4vM · 2021-11-02

**Correctness:** 3
**Technical Novelty And Significance:** 2
**Empirical Novelty And Significance:** 2
**Recommendation:** 5
**Confidence:** 4

**Main Review:**

The proposed one-sided MVI condition is interesting, and the analysis under this condition provides new insights into the convergence of adaptive optimization algorithms on min-max problems such as GANs. The theoretical analysis is clear and easy to follow.

The authors argue that "it is very unlikely that there is a optimal discriminator" based on the observation that the gradient norm of discriminators remains large throughout the training process of GANs. However, it is possible there exists an optimal solution for discriminators, which the optimizer fails to converge to, and there is no convincing evidence in the paper to prove otherwise. For instance, the MVI condition seems to hold reasonably well in some scenarios, such as the DCGAN experiments presented in Appendix C. Furthermore, the Adam variants are proposed to simplify the analysis of Adam, therefore it is necessary to show if the variants perform similarly as Adam in practice, which is not done in the empirical study. In addition, the generative performances of different optimizers are only compared qualitatively without using metrics like FID.

There are also noticeable typos or errors throughout the paper. E.g., in Sec. 2.1, "Stochastic Gradient Descent (SGD) was originally proposed by (Goodfellow et al., 2016)". And in Sec. 5, "provide the theoretical guarantee of the one-sided convergence of Adam under one-sided MVI condition", where it should be Adam-like algorithms instead?

**Summary Of The Paper:**

This paper analyzes two variants of the Adam optimizer, and proves their convergence under either the standard MVI condition or the newly proposed one-sided MVI condition. The aforementioned Adam variants are then empirically evaluated by training GANs on MNIST, Fashion-MNIST, and CIFAR-10, demonstrating better sample quality than SGD.

**Summary Of The Review:**

This paper presented some interesting theoretical analysis of Adaptive optimization algorithms, but part of its theory and experiments are not convincing enough, and the overall writing needs to be improved.

---

### Official Review · Reviewer_wnya · 2021-11-02

**Correctness:** 2
**Technical Novelty And Significance:** 2
**Empirical Novelty And Significance:** 2
**Recommendation:** 3
**Confidence:** 5

**Main Review:**

Strengths: the paper is well-written. The one-sided MVI assumption is an interesting observation.

Weaknesses:
1. The main technical proofs follow closely to [Liu et al. 2020], expect for changing the gradient to moving-averaged gradient. AMSGrad-EG uses the same MVI assumption as in [Liu et al. 2020]. AMSGrad-EG-DRD only requires one-sided MVI assumption, but the convergence measure is also weaker (only partial gradient in terms of $x$). Please explain what is the technical contribution of your Theorem 3.1 and Theorem 3.2 when compared with [Liu et al. 2020].

2. One-sided MVI is not weaker than MVI assumption, since they cannot imply each other. However, Theorem 3.2 is a weaker convergence result since the LHS is only a partial gradient in terms of $x$.

3. For both Theorem 3.1 and Theorem 3.2, the convergence crucially relies on large minibatch. This is not practical.

4. Assumption 1 (4) requires bounded iterate. However it seems that the authors did not consider a projection in their algorithm.

5. I have a very big concern about experiments. (i). The evaluation is not comprehensive and not convincing. For example, the authors only show the results on MNIST dataset which is too small for GAN. Also they only provided subjective generated picture without providing quantitative results (e.g., Inception Score and Frechet Inception Distance). (ii). The experiments presented in Figure 3 are very poor. For example, Columns (d) (e) (f) are all pictures of visually bad quality, I personally do not think column (e) is better than (d) and (f) at all.






**Summary Of The Paper:**

This manuscript developed several algorithms (e.g., AMSGrad-EG, AMSGrad-EG-DRD) for nonconvex-nonconcave min-max optimization. The convergence result of AMSGrad-EG-DRD is shown under the one-sided MVI condition. Polynomial-time complexity results are established. Some toy experiments are conducted for GAN on MNIST and fashion-MNIST datasets.

**Summary Of The Review:**

This manuscript proposed an interesting one-sided MVI assumption for nonconvex-nonconcave minimax problems, and developed some algorithms with polynomial-time complexity to first-order stationary points. The theory requires certain unrealistic assumptions (e.g., large minibatch, bounded iterate), and overall is not surprising. Empirical results are poorly evaluated and are not convincing.

---

### Official Review · Reviewer_nfde · 2021-11-03

**Correctness:** 3
**Technical Novelty And Significance:** 4
**Empirical Novelty And Significance:** 4
**Recommendation:** 6
**Confidence:** 3

**Main Review:**

This paper looks particularly interesting to me because of its clarity in presentations and its novelty in the theoretical results. As I am not an expert in min-max optimization or GANs, I cannot be very confident that the results are completely new. However, analyzing Adam-type algorithms in the training of GANs is, as far as I am aware, an important yet not sufficiently explored direction. The authors have chosen a general nonconvex, nonconcave problem to analyze, and have shown the results under the standard MVI assumption and the one-sided MVI assumption. The one-sided convergence behavior is also supported by the experiments. Therefore, I like the results of this paper.

In terms of weaknesses, I would like to discuss the following questions with the authors.

1. In terms of the assumptions in Table 1, since the authors not only need MVI but also some other assumptions (e.g. Lipchitzness and boundedness in Assumption 1), why do the authors only list MVI in the table? I spent some time reading the other papers mentioned in this table, and it seems to me that different papers have very different assumptions, and Assumption 1 does not seem to be used in all of them. Therefore, I don't think Table 1 is clear enough. I wish the authors could explain Table 1 a little more thoroughly.

2. In terms of the optimization algorithm, I wonder whether the other adaptive algorithms could also obtain similar results. Of course, AMSGrad is a perfectly fine choice. However, I don't see the technical difficulty if the authors just use a general non-decreasing $H_t$. Since the authors are claiming the convergence results for Adam-type algorithms, a general choice of $H_t$ would make their argument stronger. Besides, is it possible to use Popov's Extra gradient to reduce the number of queries to the SFO?

3. The experiments part (section 4) looks less convincing to me, or even unnecessary because the main point of this paper is on the theoretical side and to show that one-sided convergence exists. I don't see why it is needed to compare the performance of the proposed algorithm with SGDA. Also, there are no FID scores / Inception scores and it's really hard to tell which images are better.

Again, I am not an expert in this area and I might be missing something. I look forward to the authors' discussions.


**Summary Of The Paper:**

This paper analyzes the performance of Adam-type algorithms (AMSGrad, to be specific) in nonconvex nonconcave minimax optimization. The authors propose that Adam-type algorithms can converge to a stationary point with the standard MVI assumption and an even weaker one-sided MVI assumption. The authors verify their claims using Experiments.

**Summary Of The Review:**

This paper provides a solid contribution to the area of min-max optimization, with their theory supported by experiments in training GANs.

---

### Official Review · Reviewer_jDRv · 2021-11-07

**Correctness:** 3
**Technical Novelty And Significance:** 2
**Empirical Novelty And Significance:** 2
**Recommendation:** 5
**Confidence:** 4

**Main Review:**

Pros:
- The paper is well written and easy to follow.
- The convergence analysis of the proposed algorithms are solid.

Cons:
My main concerns of the paper focus on the motivation and the experiments.
- When training the GAN models, we usually require that the loss function given by the discriminator approaches to zero, in terms of either KL divergence or Wasserstein distance. Although Fig. 1 gives the gradient information of both the generator and discriminator, it is hard to say if the model really converges well. As a nonconvex problem, it is possible that the gradient of the generator approaches to zero, while the model actually doesn't converge. In such situation, the generated data may be meaningless. Moreover, with the low quality of the generated images, it is highly possible that the GAN model does not converge. Thus, I am not sure if GAN model is a proper application of the proposed algorithms.
- In the experiments, the authors use the Wasserstein GAN [Arjovsky et al., 2017] to conduct the experiments. The vanilla WGAN [Arjovsky et al., 2017] model may fail to converge due to the lack of the 1-Lipschitz constraint of the discriminator. The generator only works well with a promising and working discriminator. This means that the used WGAN model may not satisfy Assumption 1 or Assumption 3. In such situation, it is hard to differentiate if the proposed mechanism or the nonworking discriminator causes the zero gradient problem of the generator.
- To make the experiments more convincing, I suggest the authors to experiment on 2-dimensional toy dataset first, which will be much simpler and be easier to find if the model does converge.

**Summary Of The Paper:**

The paper observes the one-sided convergence phenomenon of GAN's training, and proposes the one-sided MVI condition suitable for this problem. Then the convergence analysis is provided for the proposed AMSGRAD-EG and AMSGrad-EG-DRD algorithms.

**Summary Of The Review:**

It seems that the theoretical part of the paper does not support the motivation well. If this problem can be solved, I'll raise my rating.

---

### Decision · Program_Chairs · 2022-01-20

**Decision:**

Reject

**Comment:**

This paper studies the convergence of Adam-type algorithms (two variants of AMSGrad in particular) in min-max problems that satisfy a one-sided "Minty variational inequality" condition.

The reviewers identified several weaknesses in the paper and the authors did not provide a rebuttal to these concerns so there was consensus to reject the paper.